# An inverse agonist of orphan receptor GPR61 acts by a G protein-competitive allosteric mechanism

Joshua A. Lees[1,5], João M. Dias [1,5], Francis Rajamohan[1], Jean-Philippe Fortin[2], Rebecca O'Connor[1], Jimmy X. Kong[2], Emily A. G. Hughes [2], Ethan L. Fisher [3], Jamison B. Tuttle[4], Gabrielle Lovett[4], Bethany L. Kormos [4], Rayomand J. Unwalla[4], Lei Zhang[4], Anne-Marie Dechert Schmitt[3], Dahui Zhou[3], Michael Moran [3], Kimberly A. Stevens[2], Kimberly F. Fennell[1], Alison E. Varghese[1], Andrew Maxwell[1], Emmaline E. Cote[1], Yuan Zhang[4] & Seungil Han [1] ✉

GPR61 is an orphan GPCR related to biogenic amine receptors. Its association with phenotypes relating to appetite makes it of interest as a druggable target to treat disorders of metabolism and body weight, such as obesity and cachexia. To date, the lack of structural information or a known biological ligand or tool compound has hindered comprehensive efforts to study GPR61 structure and function. Here, we report a structural characterization of GPR61, in both its active-like complex with heterotrimeric G protein and in its inactive state. Moreover, we report the discovery of a potent and selective small-molecule inverse agonist against GPR61 and structural elucidation of its allosteric binding site and mode of action. These findings offer mechanistic insights into an orphan GPCR while providing both a structural framework and tool compound to support further studies of GPR61 function and modulation.

G protein-coupled receptors (GPCRs) form one of the largest and most important classes of therapeutically relevant proteins in humans, accounting for an estimated 30-50% of the targets of currently marketed drugs[1,2]. Nevertheless, a sizable fraction of disease-relevant GPCRs, particularly among orphan receptors, have not yet been harnessed for therapeutic modulation, though many are the focus of active research. GPR61, an orphan class A (rhodopsin family) receptor closely related to biogenic amine receptors[3,4], is predominantly expressed in the pituitary and appetite-regulating centers of the hypothalamus and brainstem. Mutagenesis and human genome-wide association studies have linked GPR61 to phenotypes associated with type 2 diabetes and body mass index[5,6], making it a potential target for the modulation of appetite and body weight.

Like many other class A GPCRs[7], GPR61 is a constitutively active receptor that signals through Gαs to activate production of the small molecule second messenger cyclic AMP (cAMP) by adenylyl cyclase[8]. The mechanism underlying its constitutive activation remains poorly understood, but mutagenesis studies have suggested that residues near its N-terminus may play a key role[9]. The lack of structural information has been a significant impediment to progress in characterizing GPR61, in part because the absence of a known biological ligand or tool compound has made structural efforts challenging. While ethanolamine plasmalogens have been proposed as endogenous GPR61 ligands[10], further study is needed to better understand this activity. Furthermore, while 5-nonyloxy-tryptamine (5-NOT) has been reported as a GPR61 inverse agonist[11,12], its low potency, low solubility, and lack of selectivity have limited its utility as a tool compound for

[1]Discovery Sciences, Medicine Design, Pfizer Inc., Groton, CT, USA. [2]Internal Medicine Research Unit, Pfizer Inc., Cambridge, MA, USA. [3]Internal Medicine, Medicine Design, Pfizer Inc., Groton, CT, USA. [4]Internal Medicine, Medicine Design, Pfizer Inc., Cambridge, MA, USA. [5]These authors contributed equally: Joshua A. Lees, João M. Dias. ✉e-mail: seungil.han@pfizer.com

pharmacological or structural studies. GPCR structure determination in the absence of a ligand or tool compound is fraught with challenges, including poor protein expression and solubility, often compounded by inherent structural plasticity that can be prohibitive for high-resolution structures. This is reflected in a relative paucity of unliganded GPCR structures in the Protein Data Bank (of 524 GPCR structures solved by cryo-EM representing 130 receptors, as reported by the GPCR database[13], only 55 structures of 37 receptors are without a ligand).

Here, we report the unliganded structure of GPR61 in its active, G protein-coupled state, using cryo-electron microscopy (cryo-EM), which sheds light on the basis of its constitutive activity. GPR61 knockout mice exhibit a hyperphagic phenotype leading to obesity[14], suggesting that GPR61 inhibition by an inverse agonist could be used to treat wasting disorders, such as cachexia. Through efforts to develop a small molecule therapeutic targeting GPR61 for the treatment of cachexia, we have discovered a sulfonamide inverse agonist tool compound that exhibits potent and selective inhibition of GPR61 constitutive activity. Structural co-elucidation of GPR61 with this compound reveals it to act through an allosteric pocket by an unusual

mechanism, blocking G protein activation by binding and remodeling an intracellular pocket that is normally occupied by Gαs in the activated state. Collectively, the discoveries reported here shed light on the mechanism of GPR61 activation and a heretofore undescribed mechanism of GPCR inactivation, while providing a structural platform for future studies of GPR61 and a tool compound to support future mechanistic studies and drug discovery efforts.

## Results

### The structure of GPR61 in its active state suggests a basis for its constitutive activity

To better understand the molecular determinants of constitutive GPR61 activation, we determined its structure in complex with the Gαs/β1/γ2 heterotrimeric G protein by cryo-EM at 3.5 Å nominal resolution (Fig. 1a). Each of the components of the complex was clearly resolved, with an overall architecture consistent with that of related active-state class A receptor structures[15–18], suggesting the receptor was captured in an active-like conformation, though the binding of an agonist ligand could potentially induce further conformational change.

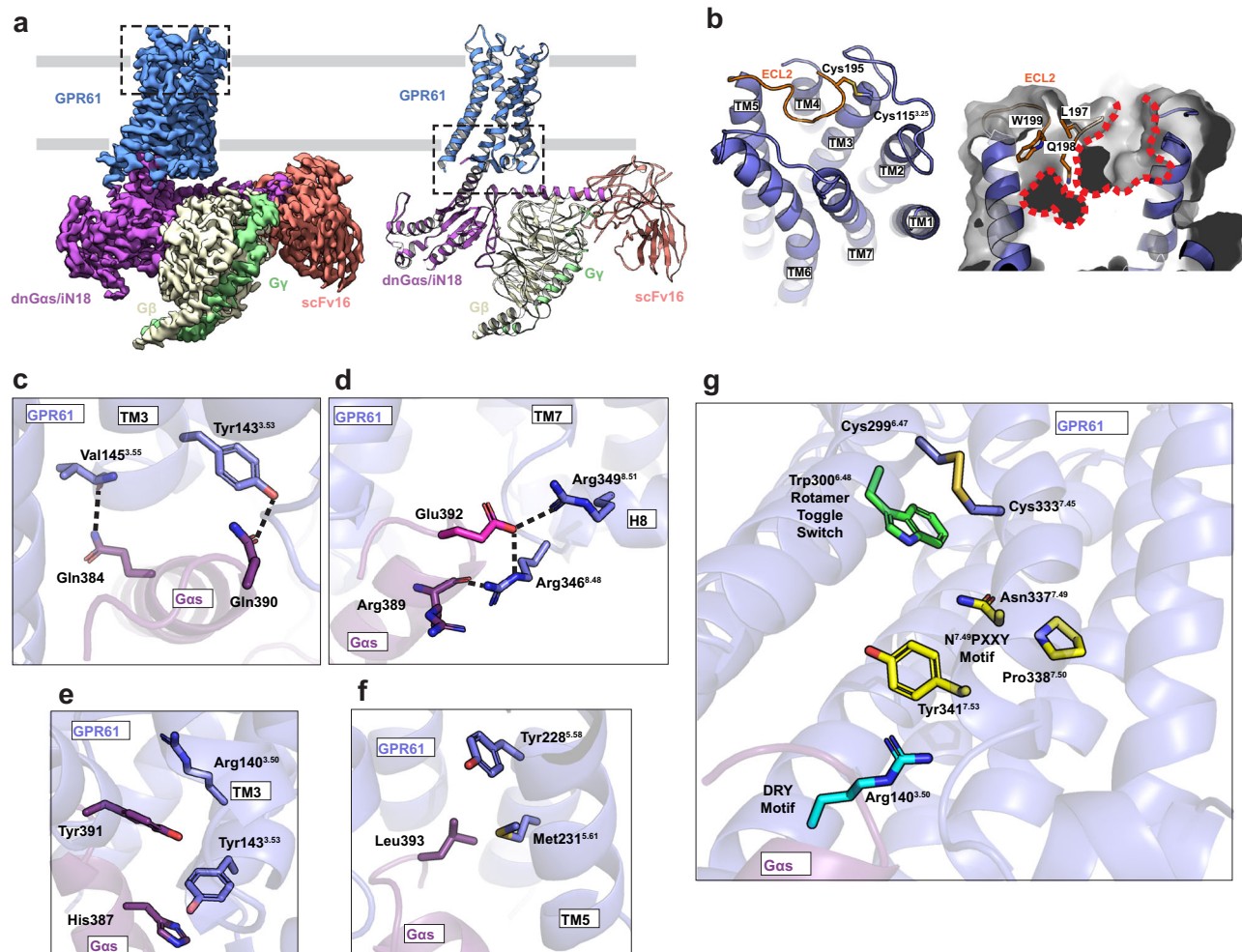

**Fig. 1 | Cryo-EM structure of active-state GPR61-G protein complex. a** Map (*left*) and model (*right*) of active-state GPR61-G protein complex with scFv16, colored by subunit. Grey lines indicate the position of the plasma membrane. **b** Highlighted by inset from **a**, left panel. *Left panel*, Extracellular face of GPR61, with ECL2 (extracellular loop 2) highlighted in orange and residues participating in the conserved disulfide shown in stick representation. *Right panel*, The orthosteric pocket of GPR61 is highlighted with a red dotted line. Key residues of ECL2 (orange) blocking access to the lower portion of the pocket are shown in stick representation. Transmembrane helices are labeled with numbered TM notations. **c–f.** Key residues involved in GPR61-Gαs interaction, corresponding to the highlighted region of A, right panel. **c** Polar interactions **d** Hydrogen bond network underlying selectivity for Gαs. **e** D/ERY motif hydrophobic stacking interactions with Gαs. **f** Hydrophobic interactions. **g** A disulfide bridging GPR61 TM6 and TM7, with nearby motifs involved in activation switching highlighted.

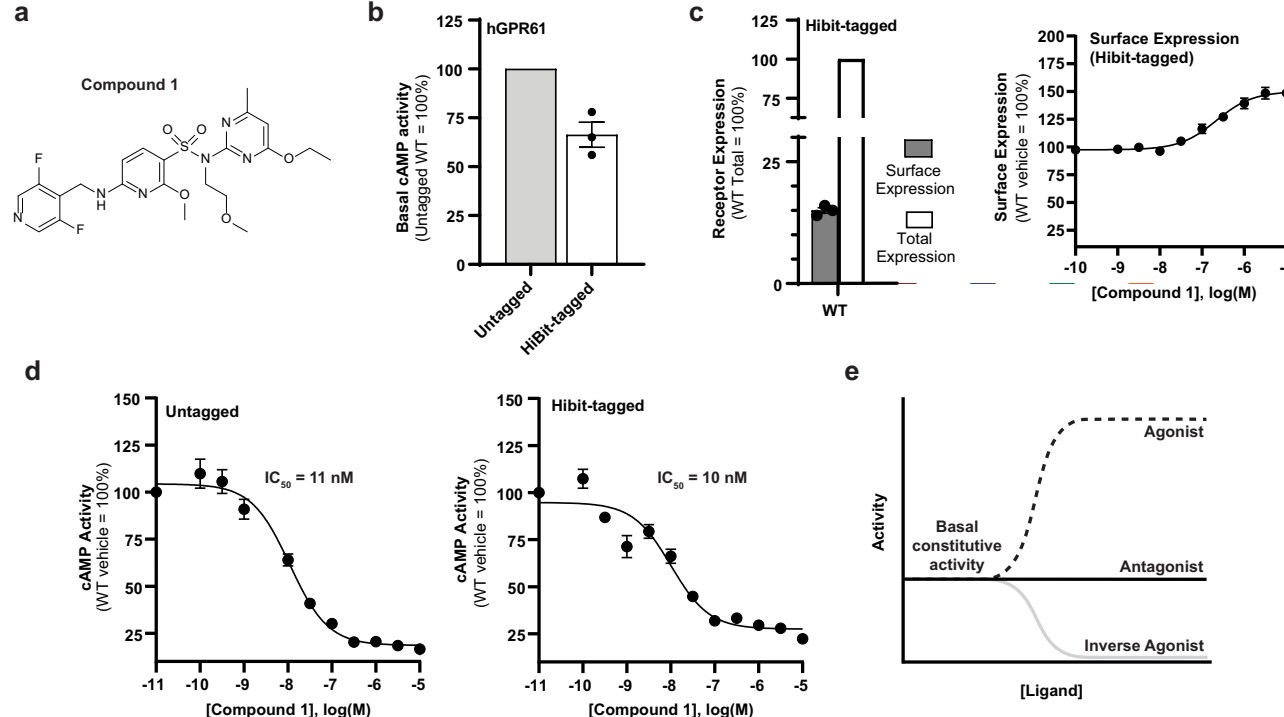

**Fig. 2 | Inverse agonist compound structure and characterization. a** Chemical structure of Compound 1. **b** Relative activity in the cAMP assay of wild-type (WT) GPR61 and HiBit-tagged GPR61 used for measuring cell surface expression. **c** Total and surface expression of GPR61 as measured using HiBit-tagged GPR61. **d** Compound 1 cyclic adenosine monophosphate (cAMP) assay inhibition curves and $IC_{50}$ values for GPR61 (Untagged and HiBit-tagged). **e** General schematic indicating the expected directions of concentration-dependent responses to agonist, antagonist, and inverse agonist ligands, respectively, by a generic constitutively active receptor. In panels **b**–**d**, bar plots and error bars represent the mean ± SEM. $N = 3$ independent experiments. Source data are provided as a Source Data file.

Much like other active class A GPCR structures[15,18,19], extracellular loop (ECL) 2 of GPR61 adopts a lid-like conformation over an orthosteric pocket, with its conformation stabilized by a conserved disulfide bond between $Cys115^{3.25}$ (residue superscripts used throughout the manuscript indicate Ballesteros and Weinstein standardized GPCR residue position notations[20], formatted with the TM helix number and associated standardized residue number separated by a period) and Cys195 of ECL2. The conformation of this loop largely occludes the orthosteric pocket, leaving a comparatively small and discontinuous pocket framed by TM1, 2, 3, and 7 and bisected by ECL2 (Fig. 1b). By comparison to other biogenic amine receptors, GPR61 lacks key ligand-interacting residues common to the orthosteric pockets of these receptors (e.g. $Asp^{3.32}$, $Asn^{6.55}$), though it retains conserved residue $Phe304^{6.52}$. Diffuse density for the extracellular portion of TM1 suggests the pocket is plastic, and displacement of ECL2 by a ligand, as for instance in the case of rhodopsin[21,22], could expose and/or frame a larger and deeper pocket. Thus, we cannot predict how the conformation of ECL2 might differ from the observed structure in the presence of a potential agonist.

GPCR signaling through the cAMP pathway relies on binding of the C-terminal helix of Gα to an exposed intracellular pocket of the receptor formed primarily by activating movements of TM5 and TM6. This binding event triggers Gα to exchange GDP to GTP, leading to its activation. In the GPR61 structure, the C-terminal helix of Gαs binds within this pocket through a network of polar contacts involving residues of TM3, TM6, helix 8, and ICL2. On TM3, $Tyr143^{3.53}$ and $Val145^{3.55}$ form stabilizing hydrogen bonds with Gαs residues Gln390 and Gln384, respectively (Fig. 1c). Key to GPR61's selectivity for Gαs are the interactions made by Gαs residue Glu392 with $Arg346^{8.48}$ and $Arg349^{8.51}$. $Arg346^{8.48}$ makes a further interaction with the mainchain carbonyl of Gαs Arg389, forming a small hydrogen bonding network (Fig. 1d). Because Glu392 is found only in Gαs and $G_{olf\alpha}$, this hydrogen

bonding pattern cannot be formed with other Gα proteins[23]. A number of non-polar interactions also drive interaction. A key residue of the D/ERY motif, $Arg140^{3.50}$, forms an intricate ladder of hydrophobic interactions with $Tyr143^{3.53}$ and Gαs residues Tyr391 and His387 (Fig. 1e). Though GPR61 lacks an ionic lock, $Arg^{3.50}$ undergoes significant conformational rearrangement during activation of related receptors[24], and this is also expected to be true of GPR61. $Tyr228^{5.58}$ and $Met231^{5.61}$ form further hydrophobic interactions with Leu393 of Gαs in the intracellular pocket formed by TM4, 5, and 6 (Fig. 1f).

The constitutive activity of GPR61 has been attributed to residues near its N terminus, with Val19 proposed to play a key role[9]. Some portion of this peptide might be expected to bind to the extracellular surface of the receptor, likely the orthosteric site, to accomplish activation. In our structure, the receptor's initial 44 residues are unresolved, suggesting that any interactions of N-terminal residues with the receptor's extracellular surface or orthosteric site are likely transient, though this does not preclude a role in activation. Efforts to capture this N-terminal region through crosslinking or addition of peptides derived from its sequence were unsuccessful. The bias of GPR61 toward constitutive activation may not rely solely on its N-terminal peptide, however, as the following key sequence and structural features could be consistent with partial destabilization of the inactive state. For instance, in some (though not all) class A receptors, $Arg^{3.50}$ of the conserved D/ERY motif participates in a salt bridge, called the "ionic lock", with an acidic residue (Asp or Glu) in position 6.30, which creates an energetic barrier to the movement of TM6 to accommodate binding of the Gα C terminus during activation[25]. In GPR61, $Glu^{6.30}$ is substituted with a glycine (disordered in this structure), preventing the D/ERY motif ($Glu139^{3.49}$, $Arg140^{3.50}$, and $Tyr141^{3.51}$) from making this favorable interaction. Similarly, binding of Na+ ions to some class A GPCRs acts to stabilize the inactive unliganded state[26–28], but GPR61 lacks key sodium-interacting residues ($Ser^{3.39}$, $Asn^{7.45}$, $Ser^{7.46}$) and thus is

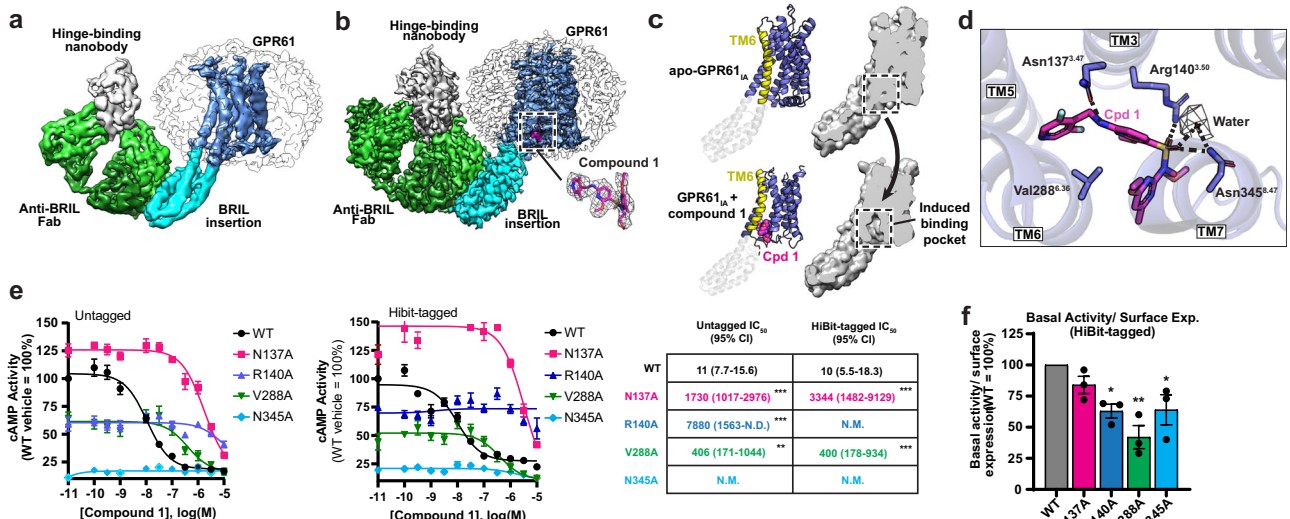

**Fig. 3 | Structural and functional analysis of compound 1 binding to GPR61$_{IA}$.**
**a** Cryo-EM map of apo GPR61$_{IA}$, colored by subunit. The sequence inserted into GPR61 (comprising BRIL and A$_{2A}$R-derived linker sequences) is colored in cyan. **b** Cryo-EM map of GPR61$_{IA}$ bound to compound 1, colored as in **a**. The compound 1 binding site is indicated by the dotted box. Inset shows compound 1, colored in magenta, fitted into its corresponding map density. **c** A comparison of apo and compound 1-bound GPR61$_{IA}$ conformations, showing the conformational changes induced by binding of compound 1. Ribbon diagrams, with TM6 highlighted, are shown at left, with a cutaway of the corresponding surface representation shown to the right. **d** Compound 1 (Cpd 1) is shown in magenta with its binding site, with key interaction residues shown in stick representation. Map density for an ordered water is shown as dark gray mesh. Hydrogen bonds are indicated by dashed lines. **e** GPR61 cAMP IC$_{50}$ curves and values for WT GPR61 and the indicated mutants. Parenthetical values in the table represent 95% CI and asterisks indicate statistical significance. **f** Basal activity of GPR61 WT and mutants normalized to relative surface expression (total activity and expression data are included in Supplementary Fig. 7a, b). Bar plots and error bars represent the mean ± SEM. In panels **e**, **f**, statistical significance is indicated with asterisks (*$p \le 0.05$, **$p \le 0.01$, ***$p \le 0.001$) and was assessed using one-way ANOVA with one-sided Dunnett's post hoc test. $N = 3$ independent experiments. Source data are provided as a Source Data file.

not susceptible to this type of negative allosteric modulation. Interestingly, the defunct sodium binding site is juxtaposed with a disulfide bond between Cys299[6.47] and Cys333[7.45] that appears to be unique among GPCRs with known structures (Fig. 1g). A pair of cysteine residues in corresponding positions is found in only one other receptor, the class A P2Y purinoceptor 10, for which no structure is known, and AlphaFold predictions of both proteins fail to predict this disulfide. Other key features associated with activating conformational changes in GPCRs, including the Trp[6.48] rotamer toggle switch and the N[7.49]PXXY motif[29], are proximal to this disulfide, which could potentially constrain the relative movements of TM6 and TM7. To probe the function of this disulfide, we generated a GPR61 mutant (C299S) unable to form a disulfide bond, but found that it exhibits a similar level of basal activity to the wild-type receptor (Supplementary Fig. 1), suggesting that it may not play a significant role in basal activation.

An agonist-like motif (ALM) found in ECL2 has recently been identified in several receptors, including GPR52, GPR21, and GPR12[30–33] as a stimulator of constitutive receptor activity. To probe the potential role of ECL2 in GPR61 activation, we examined the basal activity of GPR61 with point mutations in ECL2 designed to weaken its interactions with the orthosteric site (Q198A, W199A) and/or abolish the conserved disulfide (Δ195-201(GGSGGSGG), C195A) (Supplementary Fig. 1). While mutants that abolished the disulfide modestly decreased constitutive activity, none of the mutants caused the catastrophic loss of activity observed for similar mutants of GPR52, GPR21, and GPR12, suggesting that ECL2 is not a major contributor to GPR61's constitutive activity.

### Discovery of a potent and selective GPR61 inverse agonist
With the aim of identifying GPR61 inhibitors to treat cachexia, we initiated a high-throughput screening campaign leveraging Pfizer's internal compound libraries against an assay measuring cAMP levels in a cell line overexpressing GPR61. Initial hits emerging from this screen were extensively optimized to yield a class of potent and selective

sulfonamide-based GPR61 inverse agonists, represented here by compound 1, a tertiary sulfonamide (Fig. 2a; Supplementary Note 1, Supplementary Fig. 2). Consistent with a role in suppressing constitutive receptor activity, compound 1 caused increased cell surface expression of a HiBit-tagged form of wild-type GPR61 (Fig. 2b, c). We attribute this to compensatory overexpression in response its inhibition, though we cannot exclude the possibility that a different cause, such as pharmacochaperone activity, might account for increased surface expression of the receptor. Compound 1 demonstrates excellent potency in the functional cAMP assay (IC$_{50}$ = 10–11 nM) (Fig. 2d), with an inhibition profile consistent with that of a generic inverse agonist (Fig. 2e), and is selective for GPR61 among a panel of GPCRs (off-target IC$_{50}$ values > 10 μM), with an excellent in vitro off-target profile (Supplementary Table 1, 2). To better understand the molecular basis of Compound 1's activity, we next pursued structural studies of GPR61 in its inactive state.

### An AlphaFold-driven approach to inactive-state GPCR construct design
GPCR inverse agonists inhibit signaling by diverse mechanisms, acting at any of several allosteric sites on the receptor. Structural characterization is critical to understanding this process, so our next step was to pursue an inverse agonist-bound cryo-EM structure of GPR61. Structural characterization of inactive-state GPCRs by cryo-EM is made considerably more challenging by the loss of the heterotrimeric G protein, which acted as a fiducial for particle alignment in our active-state structure. To compensate for the loss of the G protein, we employed a strategy similar to that previously used to determine the structure of inactive Frizzled 5[34], in which thermostabilized *E. coli* apocytochrome b562 RIL (BRIL[35]) was rigidly fused between TM5 and TM6, replacing intracellular loop 3 (ICL3). Because such dual helical fusions require careful optimization to ensure continuous helicity at both junctions, we combined knowledge gleaned from both our active-state GPR61 structure and the published Frizzled 5 structure to design

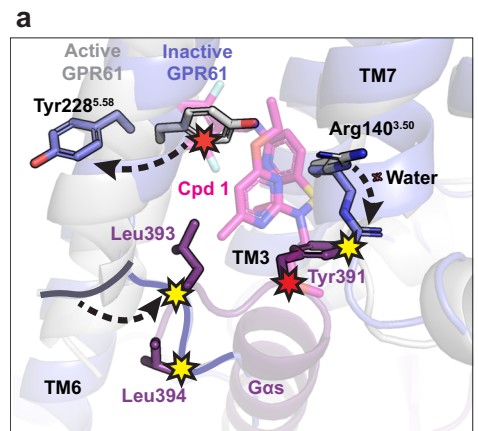

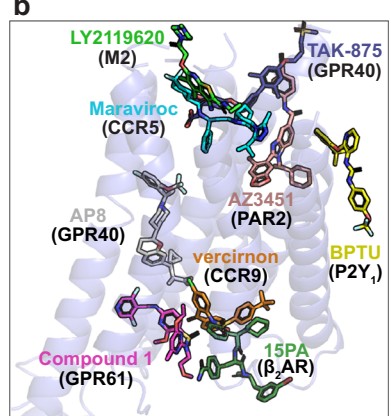

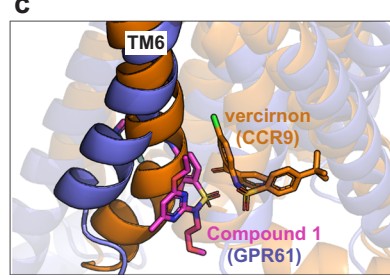

**Fig. 4 | Analysis of compound 1 inverse agonist mechanism. a** Key residue clashes and conformational changes induced by binding of compound 1 (Cpd 1) to GPR61. The structure of active GPR61 (light grey) is overlaid with the compound 1 (magenta)-bound structure of inactive GPR61 (blue), with key residues highlighted in stick representation. Clashes with the compound are indicated by red stars, while clashes with Gαs induced by compound binding are indicated by yellow stars. **b** Compound 1 defines an unusual allosteric site and mechanism. The structure of compound 1-bound GPR61_IA is shown in ribbon representation, with published exemplars[45,62–68] representing the known allosteric sites of class A GPCRs superimposed, colored as indicated. **c** Compound 1-bound GPR61_IA and vercirnon-bound CCR9[45] structures, colored as indicated, are superimposed. Vercirnon occupies the known allosteric site that lies nearest to that of compound 1. The different conformations of TM6 induced by these two inverse agonists are highlighted.

a series of 25 GPR61-BRIL fusion constructs and screen their purified protein products by cryo-EM. A promising early construct yielded a model resolved to approximately 6 Å with clear separation of the seven TM helices, but efforts to improve the resolution of this construct further were unsuccessful. The subsequent availability of AlphaFold 2 allowed us to retrospectively compare the cryo-EM screening results for the fusion constructs against their corresponding AlphaFold predictions. We observed that constructs whose predictions showed rigid helical fusions with high confidence correlated with increased order in the results of our cryo-EM evaluation, building confidence in Alpha-Fold 2 as a tool for improved construct design. We thus employed an AlphaFold-first strategy to screen new construct designs in silico (Supplementary Fig. 3), selecting a subset of four for further characterization by cryo-EM. AlphaFold predictions for all four indicated high-confidence helicity at the junctions between TM5/6 and the BRIL helices, but one construct, designated GPR61_IA, stood out clearly as the best-ordered after screening by cryo-EM. While functional analysis of this construct was not possible with the BRIL fusion, a comparison of the optimized construct's expression at the cell surface against that of the wild-type receptor indicated that it is properly trafficked to the cell surface and thus well-folded (Supplementary Fig. 4). This construct was used for all subsequent cryo-EM structural studies in combination with a previously described BRIL-binding Fab and a hinge-stabilizing nanobody[34,36] as fiducials.

## Compound 1 binds an induced allosteric site in GPR61

The structure of apo-GPR61_IA was determined by cryo-EM to a nominal global resolution of 3.97 Å (Fig. 3a), but the region corresponding to the receptor was poorly resolved. While this map was sufficient to identify and flexibly fit the TM helices to the density, side chains could

not be fitted confidently. As expected, the positions of the TM helices revealed significant structural differences compared to the active-state receptor structure solved with the Gαs protein complex. The most pronounced of these was an inward rotation of the intracellular half of TM6 by about 12° toward TM1, 2, 3, and 7 relative to the active-state structure. As expected for an inactive-state GPCR, this movement would cause TM6 to clash with the position of the C-terminal helix of GαS, preventing G protein complex binding and activation.

To obtain insights into the inhibitory mechanism of our inverse agonist, we pursued a cryo-EM co-structure of GPR61_IA in the presence of compound 1. The presence of the inhibitor significantly improved the order of the receptor relative to the apo structure, and the final map was resolved to 2.9 Å with excellent local resolution for the receptor (Fig. 3b). Modeling of the receptor into this map revealed a well-resolved region of unmodeled density whose shape is congruent to compound 1 (Fig. 3b, inset). Unexpectedly, compound 1 binds an induced allosteric pocket situated on the intracellular side of the receptor and flanked by TM helices 3, 5, 6, and 7. The formation of this induced binding pocket is enabled by a counter-intuitive conformational change in which the intracellular half of TM6 is forced outward relative to its position in the apo structure, more closely resembling the active than the inactive form of the receptor (Fig. 3c), though the detailed interactions differ significantly from the active state. Alignment of this structure to the receptor's active state gives an overall RMSD of 3.47 Å across the receptor, but an RMSD of only 0.44 Å for TM6. To accommodate this repositioning, the helical linkage between TM6 and BRIL is disrupted, with TM6 residues up to Lys284[6.32] becoming disordered.

Key features of compound 1's induced binding pocket reveal the basis of its potency. The bound conformation of compound 1 wraps

around the side chain of Val288[6.36], forming extensive stabilizing Van der Waals' contacts (Fig. 3d). Its difluoropyridine group projects into a hydrophobic gap between TM5 and TM6, while the central linker's methoxypyridyl is flanked by hydrophobic interactions with Val288[6.36] and the β and γ carbons of Arg140[3.50]. The terminal methylpyrimidine projects toward the surrounding micelle by sandwiching between helices 6 and 7, while its ethoxy group extends toward Tyr341[7.53] of the NPxxY motif (Supplementary Fig. 5).

Most critical for potency is compound 1's sulfonamide moiety. Sulfonamides constitute a privileged chemotype among GPCR modulators, with many published examples[37–40]. The unique allosteric site bound by compound 1, however, defines a class of sulfonamide GPCR inhibitors. The sulfonamide oxygens of compound 1 form key hydrogen bonds with Asn345[8.47] and Arg140[3.50], the key residue of the widely conserved D/ERY motif associated with activating conformational changes (Fig. 3d). Strong density for an ordered water is discernable in the map, coordinated by Asn345[8.47] and the sulfonamide. Mutation of R140 or V288 to alanine made the receptor less sensitive to inverse agonism by compound 1 in the cAMP assay, while changing constitutive activity by only about 2-fold (Fig. 3e, f). In contrast, an N345A mutation significantly reduced the basal cAMP activity of the receptor, but additional investigation revealed this mutation to reduce the fraction of GPR61 at the plasma membrane (Supplementary Fig. 6). This may be attributable to the intracellular, solvent-exposed position of N345, whose mutation may impact receptor trafficking to the plasma membrane through the secretory pathway. When the N345A mutant's basal activity was normalized to its cell surface expression, its activity was similar to that of the other mutants (Fig. 3f), but showed no sensitivity to compound 1 at up to 10 μM concentration.

The similarity of compound 1's GPR61-bound conformation to its global energetic minimum conformation likely also contributes to its potency. The strain energy of compound 1's GPR61-bound conformation compared to the global minimum conformation is fairly small, estimated at ~7.0 kcal/mol, with relatively small conformational differences. (Supplementary Fig. 7a, c). Torsional energy scans of the most disparate dihedral angles between the two conformations suggest very little strain associated with the adaptation of the difluoropyridyl tail to the binding pocket (Supplementary Table 3 and Supplementary Fig. 7a), but slightly larger strain energies are required for the amine and sulfonamide torsions that lead to the bound conformation (Supplementary Table 3 and Supplementary Fig. 7b, c). Since the overall strain energy is less than those of the individual torsion profile energy differences, the individual torsion scans likely overestimate the strain energy.

Compound 1 was found to be selective for GPR61 in a panel of common off-target GPCRs and against related receptors GPR62 and GPR101 (Supplementary Table 1, 2; Supplementary Fig. 8). Its selectivity may be attributed to the characteristics of the allosteric pocket. Compound 1 makes a key hydrogen bond through its secondary benzylic amine to the terminal amide oxygen of Asn137[3.47] (Fig. 4a). The asparagine in this position is unique among known GPCRs, and in other receptors, substitutions in this position are non-conservative, with Ala and Ser being the most common replacements (Supplementary Fig. 8). As a key contributor to the compound's potency, mutation of Asn137 would be expected to exact a large energetic penalty, reducing the compound binding affinity considerably. Consistent with this hypothesis, mutation of Asn137[3.47] to Ala in GPR61 reduced the potency of compound 1 about 100-fold relative to wild-type in the cAMP assay (Fig. 3e), making it a key contributor to compound 1's selectivity. In contrast, the residues interacting with the sulfonamide moiety lend potency, but are much more highly conserved and thus do not contribute significantly to selectivity. The remainder of the hydrophobic pocket is poorly conserved among other receptors, and sequence variation in these residues would also be expected to alter the pocket's

shape complementarity to the compound for Van der Waals interactions, reducing its affinity to varying degrees.

## A G protein-competitive inverse agonist mechanism

The induced binding pocket occupied by compound 1 gives rise to an unusual mechanism of GPCR inverse agonism. As discussed above, when bound to compound 1, TM6 of the receptor adopts a conformation resembling that of the active receptor, but that nonetheless precludes the binding of Gαs necessary for activation of downstream signaling. Compound 1 acts as a "wedge", binding in a pocket that partially overlaps that bound by Gαs in the active state. This wedge pushes TM6 outward compared to its position in the inactive-state apo structure, but not quite as far as seen in the active structure (Fig. 4a). This creates subtle differences in the positioning of the other flanking helices, which remodel the Gαs-binding pocket to reposition key hydrogen-bonding residues while the methoxyethyl substituent of the tertiary sulfonamide clashes with Tyr391 of Gαs.

To accommodate the terminal pyridine moiety of compound 1, the side chain of Tyr228[5.58] on the displaced TM5 is flipped outward toward solvent, preempting a favorable hydrophobic interaction with Leu393 of Gαs. Likewise, the position of TM6 differs slightly from the active-state structure, being positioned slightly closer to the neighboring helices and creating potential clashes with residues Leu393 and Leu394 of Gαs. L393 is framed by hydrophobic interactions with residues on TM5 and TM6, both of which are shifted by the presence of compound 1 (Fig. 4a). A significant outward shift of TM7 breaks interactions with Gαs residue Tyr391, while repositioning of Arg140[3.50] disrupts the hydrophobic ladder of interactions with Gαs observed in the active-state structure and causes it to directly clash with Tyr391. Collectively, these structural changes remodel the Gαs-binding pocket to make G protein binding unfavorable. As a result, this inverse agonist mechanism paradoxically maintains a receptor conformation that broadly resembles the active state while still effectively blocking binding of the G protein.

The allosteric site described here does not overlap with any of the known GPCR allosteric sites discovered to date[41,42] (Fig. 4b). The most proximal known GPCR allosteric site is the highly conserved intracellular allosteric site observed for the β2AR, CCR, and CXCR receptors[43,44], which neighbors, but does not overlap with that of compound 1. This site is flanked by helices 1, 2, 3, 6, and 7 and, like all other known allosteric inverse agonists, stabilizes the "inward" conformation of TM6 as its means of blocking Gα association, as exemplified by the structure of CCR9 bound to the inverse agonist vercirnon, another sulfonamide (PDB: 5LWE[45]) (Fig. 4c).

## Discussion

GPR61 is an orphan class A GPCR with therapeutically relevant links to metabolic phenotypes. While the lack of structural information and tool compounds has presented challenges in studying GPR61, the active-like structure reported here provides some insights. Although assembly of the native complex by co-expression of GPR61, Gαs, Gβ1, and Gγ was unsuccessful, fusion of a dominant negative[46] variant of the Gαs/iN18 chimera[16] to the receptor's C terminus, combined with the use of a single-chain Fab (scFv16)[47], stabilized formation of the GPR61-G protein complex for structural studies. This chimera forms, with Gβ, an epitope for scFv16, which stabilized interactions between Gα and Gβ. Dominant negative Gα subunit variants have also been used successfully to enhance the formation of GPCR-Gα complexes[46,48,49]. Together, these features made structural elucidation of the complex possible. Although constitutive activation of GPR61 was suggested to involve the receptor's N terminus, the structural basis of its constitutive activation is unknown, and the possibility of a yet-to-be-identified agonist present under expression conditions has not been categorically excluded. Unexpectedly, a disulfide observed between TM6 and TM7, while adjacent to the critical tryptophan rotamer toggle

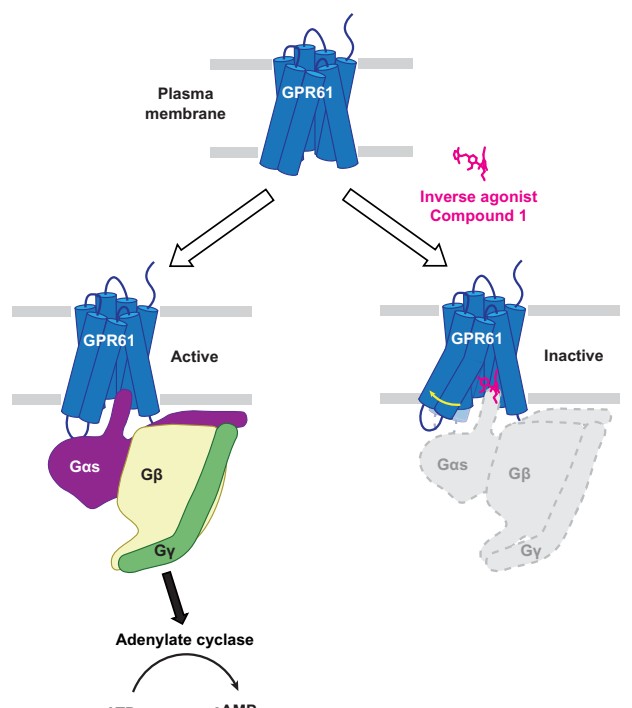

**Fig. 5 | A G protein-competitive mechanism of allosteric GPCR inverse agonism.** In its constitutively active state (top panel), GPR61 adopts a conformation that allows binding and nucleotide exchange of the G protein complex (bottom left panel) to stimulate cyclic AMP-mediated signaling through activation of adenylate cyclase. Inverse agonist compound 1 (magenta) binds to an intracellular allosteric pocket overlapping that bound by Gαs and acts as a wedge to remodel the flanking helices (yellow arrow), destroying the Gαs-binding pocket and creating direct clashes that prevent potential Gαs binding.

switch, proved not to be critical for GPR61 constitutive activity. The agonist-like motif (ALM) of ECL2 reported for GPR52, GPR21, and GPR12[30–33] to drive constitutive activity also does not appear to be present in GPR61, and we have found no structural evidence of binding to the orthosteric site by N-terminal residues with a reported role in activation[9]. The lack of some key sequence features found in many other receptors, such as an ionic lock or sodium binding site, may suggest less stabilization of the inactive state. Nonetheless, additional questions remain about the full basis of GPR61's constitutive activity.

To obtain structures of GPR61 in its inactive state, we utilized an efficient AlphaFold-driven in silico construct evaluation strategy to streamline the time-consuming process of experimental construct screening and optimization, providing significant savings in time and cost. This strategy enabled the low-resolution structure of GPR61's inactive state, revealing the conformational changes associated with its activation, as well as the basis of its constitutive activity. While AlphaFold 2 was used to screen the BRIL fusion constructs used for the inactive-state structure, a subsequent comparison of the AlphaFold 2 GPR61$_{IA}$ prediction to the inactive-state AlphaFold MultiState prediction for wild-type GPR61 revealed that they are in striking agreement, while the AlphaFold 2 prediction for wild-type GPR61 adopts a conformation intermediate between the active and inactive states. This suggests that AlphaFold2 may be a reliable tool for screening such fusions, but it is possible that improved predictions could be obtained using AlphaFold MultiState. Interestingly, however, the position of TM6 in the AlphaFold MultiState prediction for active GPR61 shows relatively poor agreement with our active-state cryo-EM structure, being positioned inward slightly, where it would clash with the position of Gαs. While the inactive apo structure's low resolution precluded a rigorous and detailed structural analysis, the shift in the

conformation of TM6 relative to the active state is similar to that observed in other inactive-state receptors, and the receptor's overall conformation is broadly consistent with the inactive conformation predicted for GPR61 in the GPCR database using AlphaFold 2-MultiState[50] and agrees well with the AlphaFold 2 prediction for the GPR61$_{IA}$ construct.

Compound 1 is a sulfonamide inverse agonist with potent and selective activity against GPR61. To better understand the molecular basis of its activity, we used the AlphaFold-designed GPR61$_{IA}$ construct to determine structure of GPR61 bound to compound 1 by cryo-EM at 2.9 Å resolution. While the artificial nature of the GPR61-BRIL fusion in GPR61$_{IA}$ may give rise to concerns about whether the conformation of the apo protein is biologically relevant, the structural rearrangement induced by compound 1 suggests that the protein derived from this construct remains sufficiently flexible to accommodate binding of compound 1 in an induced pocket. Compound 1 binds to an induced intracellular allosteric pocket that overlaps the binding site for Gαs and unexpectedly promotes a GPR61 conformation more similar to its active state than the inactive state (Fig. 5). Unusually, it acts as a competitor for Gαs binding through steric clashes and remodeling of the pocket to reduce its shape complementarity to Gαs. This stands in contrast to other known inverse agonists, which prevent Gαs binding by indirectly promoting a receptor's inactive conformation. By directly blocking binding of Gαs to GPR61, compound 1 fits the mechanistic criteria for an inverse agonist, which blocks constitutive GPCR signaling, as opposed to an antagonist, which prevents receptor activation above its constitutive level. While sulfonamides are a recurring chemotype among GPCR modulators[37–40,45,51], compound 1's inhibitory mechanism defines a separate class of sulfonamide inverse agonists not previously observed.

Together, the information we present here constitutes a toolbox for future study of GPR61 and other receptors, which we hope will enable studies of receptor function and possibly facilitate receptor de-orphanization. AlphaFold2 has rapidly gained traction as a tool for enabling structural biology, and we anticipate that AlphaFold-informed construct design strategies similar to the one reported here will be useful for helping to elucidate other GPCRs that still lack structural information. In addition to providing mechanistic information, the structures described here provide a strategic platform for future mechanistic studies of GPR61 and modulators, while Compound 1's potency and excellent off-target profile make it a high-quality tool compound for future functional studies of GPR61 in vivo and provides insights, including an unexpected allosteric pocket, that may aid future drug discovery efforts.

## Methods
### Construct design, protein expression and purification for cryo-EM study
The expression construct for human GPR61 in the active conformation (termed hGPR61-dnGαs/iN18), was designed with an HA signal peptide, FLAG tag, TEV protease cleavage site, BRIL (cytochrome b562 RIL), and PreScission protease cleavage site at the GPR61 N terminus, and with the GPR61 C terminus fused via a (GSS)$_9$ linker to a chimeric dominant negative Gαs/iN18 (combining truncation Gαi N18, Gαs N26[16] with a previously described dominant negative version of Gαs[46]).

The expression construct for human GPR61 inactive conformation (termed hGPR61$_{ICL3}$BRIL) was designed with an HA signal peptide at the N-terminus, insertion of BRIL in intracellular loop 3, and a C-terminal FLAG tag. BRIL residues were inserted between wild-type GPR61 ICL3 loop residues R233 and K285 using two short, modified linkers derived from A$_{2A}$ adenosine receptor (ARRQL between residue R233 and the N terminus of BRIL, and ERARSTLQKEV between the C terminus of BRIL and residue K285). These constructs were synthesized (Azenta), sub-cloned into pFastBac1, and expressed in *Spodoptera frugiperda* (Sf9) insect cells (Thermo Fisher) grown in SF-900

III (In) serum-free medium. These recombinant constructs were transformed into DH10Bac *E. coli* competent cells to make expression bacmid DNA, which was transfected into Sf9 insect cells to generate P0 virus following the manufacturer's (Invitrogen) instructions. Protein expression was performed by infecting 1 L of Sf-9 cells at a density of $2 \times 10^6$ viable cells/ml and an MOI of 0.5 in a serum-free insect cell medium (SF-900 III). Maximum expression of the recombinant protein was observed 48–72 h after infection and the cells were harvested when their viability was 80–85%. Harvested cells were stored at −80 °C until further use.

For purification of GPR61 constructs, frozen cell pellets were thawed in Dulbecco phosphate-buffered saline (DPBS, Lonza) and incubated for 30 min at room temperature, supplemented with EDTA-free cOmplete Protease Inhibitor (Roche) tablets, then transferred to 4 °C for another 30 min. Membranes were prepared by passing the cells through a chilled microfluidizer processor M-110L Pneumatic at 15kPsi (Microfluidics, Inc.). The lysates were centrifuged at 235,000 × g in a Type 45 Ti rotor with a Beckman Coulter Optima LE-80K ultra-centrifuge for 45 min. The membranes were resuspended and washed in 500 mM NaCl, 50 mM HEPES pH 7.5 supplemented with EDTA-free cOmplete Protease Inhibitor (Roche) tablets, and centrifuged as before for 45 min. The membranes were resuspended in a low salt buffer containing 150 mM NaCl, 50 mM HEPES pH 7.5, 10 % glycerol, supplemented with EDTA-free cOmplete Protease Inhibitor (Roche) tablets, and either frozen at −80 °C for storage or used directly. For purification, membranes were solubilized for 90 min with 1% Lauryl Maltose Neopentyl Glycol (LMNG, Anatrace) and 0.2% Cholesteryl Hemi-succinate Tris-salt (CHS, Anatrace) in 500 mM NaCl and 50 mM HEPES pH 7.5, 100 µM TCEP, supplemented with EDTA-free cOmplete Protease Inhibitor (Roche) tablets. The solubilized protein was clarified by ultra-centrifugation, as above, and the supernatant was incubated with anti-FLAG M2 (Sigma) affinity gel for 2 h at 4 °C. The FLAG resin was washed twice with a buffer containing 500 mM NaCl, 50 mM HEPES, pH 7.5, 100 µM TCEP, 0.5% LMNG and 0.01% CHS. The protein was eluted using 0.25 mg/ml FLAG peptide, 0.01% LMNG, 0.002% CHS, 150 mM NaCl, 50 mM HEPES pH 7.5, 100 µM TCEP. Eluted fractions were pooled, concentrated in a centrifugal filter concentrator, and purified by size exclusion chromatography on a Superose 6 (Cytiva) column, using a buffer containing 0.001% LMNG, 0.0002% CHS, 150 mM NaCl, 25 mM HEPES pH 7.5, 100 µM TCEP. The fractions were analyzed by SDS-PAGE and fractions corresponding to monomeric GPR61 were pooled.

To allow preparation of the heterotrimeric G protein complex, the open reading frame DNA sequences of human guanine nucleotide-binding Gβ1 corresponding to amino acids S2-N340 (UniProt P62873-1), with an N-terminal 6X His tag, and human guanine nucleotide-binding protein Gγ2 corresponding to amino acids M1-C68 (UniProt P59768-1) were synthesized and separately cloned into pFastBac1 (Thermo Fisher). For protein expression, Sf9 cells were co-infected with both Gβ1 and Gγ2 at a cell density of $2.5 \times 10^6$ cells/ml with a multiplicity of infection of 0.5 for each baculovirus. At 72 h after infection, the cells were harvested and frozen at −80 °C. Cells were lysed as described above in buffer containing 20 mM HEPES pH 8, 150 mM NaCl, 1 mM TCEP, 0.5 mM EDTA, EDTA-free protease inhibitor cocktail (Roche), and benzonase, then loaded on $2 \times 5$ ml HisTrap Crude FF (GE Healthcare) equilibrated in the same buffer. The column was washed until a stable baseline was established, then the protein was eluted in a buffer containing 20 mM HEPES pH 8, 10 mM NaCl, 1 mM TCEP, 0.5 mM EDTA, 200 mM imidazole and fractions were collected. Fractions containing the heterodimer were pooled and dialyzed overnight against 20 mM HEPES pH 8, 1 mM TCEP, 0.5 mM EDTA. The dialyzed sample was passed through a HiTrap Q column (Amersham Biosciences) equilibrated in the same buffer to bind the protein. The Gβ1γ2 heterodimer was eluted with a linear NaCl gradient in the same buffer. Fractions were analyzed by SDS-PAGE and fractions containing

the heterodimer were pooled and subjected to gel filtration on a Superdex 75 column in buffer containing 20 mM HEPES pH 8, 150 mM NaCl, 1 mM TCEP, 1 mM EDTA. Peak fractions were pooled, snap-frozen, and stored at −80 °C until use.

To prepare the hGPR61-dnGαs/iN18-G protein complex, the purified hGPR61-dNGαs/iN18 fusion protein was incubated with an excess of Gβ, Gγ, scFv16 and apyrase (New England Biolabs), for 1 h on ice following published protocols[16,47]. The complex was used directly for cryo-EM grid preparation without concentration.

For the preparation of the inverse agonist complex, the purified hGPR61$_{ICL3}$BRIL fusion was incubated on ice for one hour with an excess of anti-BRIL Fab and anti-Fab nanobody according to published protocols[34]. The complex was concentrated and purified by size exclusion chromatography using a Superose 6 Increase 5/150 GL (Cytiva) in a buffer containing 0.001% LMNG, 0.0002% CHS, 150 mM NaCl, 25 mM HEPES pH 7.5, 100 µM TCEP. The fractions corresponding to the ternary GPR61+Fab+Nb complex were pooled. The complex was incubated with 100 µM of inverse agonist compound 1 overnight and was used directly without further concentration steps for cryo-EM grid preparation.

## Cryo-EM sample preparation
Purified protein samples were subjected to centrifugation at 13,200 × g for 10 min to remove aggregates. Gold Quantifoil R1.2/1.3 200 mesh grids were made hydrophilic by glow discharge in residual air at 15 mA for 30 s using a Pelco Easiglow. In a Vitrobot Mark IV operated at 4 °C and 100% humidity, 4 µl of sample supernatant was applied to a grid, then blotted away from both sides before being vitrified by plunge-freezing in liquid ethane cooled by liquid nitrogen. Vitrified grids were stored under liquid nitrogen until imaging.

## Cryo-EM data collection and processing
Grids were imaged in a Titan Krios G2 transmission electron microscope operated at 300 kV equipped with a Falcon 4i direct electron detector and Selectris X imaging filter. All screening and data collection were performed in EPU (Thermo Fisher Scientific). Movies in EER format were collected at 215,000× magnification (0.59 Å magnified pixel size at the specimen level) with a total electron dose of 50 e⁻/Å². 

For the hGPR61-dnGαsiN18/Gβ/Gγ/scFv16 complex (Supplementary Fig. 9), a dataset of 20,635 movies was collected. Movies were subjected to patch motion correction (nominal pixel size = 0.59 Angstroms, EER fractionation into 40 frames, without upsampling) and patch CTF correction in CryoSPARC 3.3.1[52], followed by blob-based autopicking, using minimum and maximum particle diameters of 100 and 150 Angstroms, respectively. 2,768,397 particles were extracted in 600-pixel (35.4 nm) boxes Fourier-cropped to 300 pixels to give a pixel size of 1.18 Angstroms, then subjected to 2D classification in 200 classes. 2D classes showing signs of secondary structure (255,448 particles) were subjected to a second round of 2D classification into 200 classes. 188,276 particles were subjected to 3D ab initio modeling in 4 classes. The best model, comprising 52,887 particles, was subjected to non-uniform 3D gold-standard refinement and reached a final resolution of 3.47 Å, based on the FSC = 0.143 criterion.

For GPR61-BRIL fusion constructs, screening datasets of 5000 movies were initially collected and processed in CryoSPARC as described below up to 2D classification. The appearance and quality of 2D classes were used to compare and evaluate constructs. Once the final construct was selected, a dataset of 16,126 movies was collected in the presence of inverse agonist compound 1 (Supplementary Fig. 10), while a dataset of 10,000 movies was collected from the equivalent sample without compound to solve the apo structure (Supplementary Fig. 11). Movies were subjected to patch motion correction (nominal pixel size = 0.59 Angstroms, EER fractionation into 40 frames, no upsampling) and patch CTF correction in CryoSPARC 3.3.1, followed by blob-based autopicking, using minimum and maximum particle

diameters of 100 and 150 Angstroms, respectively. Particles were extracted in 500-pixel (27.6 nm) boxes Fourier-cropped to 250 pixels to give a pixel size of 1.18 Angstroms, then subjected to 2D classification into 200 classes. 2D classes resembling a micelle with protruding density were manually selected and subjected to a second round of 2D classification in 200 classes. Well-defined 2D classes representing all discernable particle views were then used for template-based picking against the dataset. Picked particles were extracted as before and subjected to two rounds of 2D classification into 200 classes. Well-resolved 2D classes were fed into 3D ab initio modeling (starting target resolution 12 Angstroms, final target resolution 5 Angstroms, 300 and 1000 particles in starting and final batch sizes, respectively), followed by non-uniform gold-standard 3D refinement with the starting ab initio model low-pass filtered to 10 Angstroms.

### Model building and refinement

For each of the GPR61 structures reported here, an atomic model predicted by AlphaFold 2[53,54] (wild-type GPR61 for the active state structure and hGPR61$_{ICL3}$BRIL fusion construct sequence for the inactive state) was rigid-body fitted into the map density. The model was successively hand-built into the map using Coot (v0.9.8.1[55]) in alternation with real-space refinement in Phenix (v1.20[56]) to produce the final model. Starting models of Fab24 BAK5 and the hinge-binding nanobody were derived from PDB entry 6WW2[34], while initial models of dnGαs/iN18, Gβ, Gγ, and scFv16 were created by modifications of PDB entry 3SN6[15]. The full cryo-EM data processing workflow and validation metrics (Supplementary Figs. 9, 10, and 11) and the model refinement statistics (Supplementary Table 4) can be found in the supplementary materials. Figures based on the structure were produced in PyMol version 2.5.4, UCSF Chimera version 1.16[57], and ChimeraX version 1.4[58].

### Computational chemistry

The strain energy of compound 1 was computed by taking the energy difference between the cryo-EM (local minimum) and global minimum conformations. The energy of the local minimum conformation was determined by minimizing the cryo-EM structure of compound 1 using a 10 kJ/mol Å$^2$ constraint on all torsions. The energy of the global minimum conformation was determined by performing a conformational search of compound 1 and selecting the lowest energy conformation. Torsional energy scans were performed by defining the dihedral angle(s) of interest in compound 1 and increasing them from 0° to 360° by increments of 10° using Coordinate Scan. All calculations were run using Macromodel[59] in Schrödinger 2021-2[60] using a dielectric constant to approximate water using default options unless otherwise specified. All conformational energies were determined using the OPLS4 force field[61], which was customized using the Force Field Builder panel for missing ligand torsion parameters.

### Cell surface expression of GPR61 WT vs. BRIL fusion (GPR61$_{IA}$)

Recombinant expression constructs for wild-type GPR61 and GPR61$_{IA}$ (both N-terminally HA-tagged) were expressed in *Spodoptera frugiperda* (Sf9) insect cells grown in SF-900 III (In) serum-free medium as described above. Protein expression was performed by infecting 1 L of Sf-9 cells at a density of $2 \times 10^6$ viable cells/ml and an MOI of 0.5 in a serum-free insect cell medium (SF-900 III). Flow cytometry was performed 42 hours post-infection using the Guava EasyCyte HT instrument and an anti-HA Alexa 488-conjugated mouse IgG antibody (R&D Systems; IC6875G). In brief, 200,000 cells from control (parental) and each version of GPR61-expressing cells at 1000 cells per microliter were distributed in triplicate to wells of a 96-well plate (Costar 3897). To wells containing each cell type, 100 μl of either working solution 1 (20% 7-aminoactinomycin D solution and 3.2% BSA in TBS), working solution 2 (working solution 1 + 2 μg/ml anti-HA Alexa 488 antibody),

or working solution 3 (working solution 2 + 0.1% Triton X-100) was added and incubated 60 min at room temperature with nutation. After incubation, 100 μl of TBS was added to each well and the plate was centrifuged for 5 min at 500 × g. Supernatants were removed and wells were washed with 200 μl TBS and centrifuged again as before. Supernatants were removed and each well was resuspended in 200 μl TBS. Cell suspensions were measured on the Guava EasyCyte.

### In vitro pharmacological analyses

**Plasmid and cell line generation.** The human GPR61 (reference sequence NM_031936.4) cDNA was synthesized and subcloned into pcDNA5/FRT/TO (Thermo Fisher, Pittsburgh, PA). The pcDNA5/FRT/TO/ human GPR61 plasmid was then transfected into the Flp-In™-CHO Cell Line (Thermo Fisher, Pittsburgh, PA) and a doxycycline-inducible clonal cell line was selected (CHO TREx hGPR61 WT). Sequences encoding untagged and HiBit/hemagglutinin (HA)-tagged versions of the wild-type human GPR61 were synthesized and cloned into pcDNA3.1. The HiBit (VSGWRLFKKIS) and HA (YPYDVPDYA) tags were sequentially included after the N-terminal initiator methionine. Four single amino acid substitutions (N137A, R140A, V288A, N345A) were then introduced in the untagged and tagged human GPR61 constructs using site-specific mutagenesis. The nucleotide sequences of all receptor constructs were confirmed by automated DNA sequencing.

**Receptor expression studies.** Parental CHO-K1 cells were maintained at 37 °C in a humidified environment (5% CO$_2$) and grown in Dulbecco's Modified Eagle Medium (DMEM) F-12 media (Gibco, 11320033) supplemented with 10% FBS, 1% Glutamax (Gibco, 35050061) and 100units/ml penicillin and streptomycin (Gibco, 15140122. Cells were dissociated using a cell dissociation solution (Gibco, 13151-014), counted, and resuspended in complete culture media without antibiotics. Cells were then transiently transfected using Fugene6 (Promega) in a 384-well plate format (Corning, 3570). For each well, 7500 cells were transfected with 5 ng of receptor plasmid, 20 ng of carrier DNA (Promega, PAE4881) and 0.075 μl Fugene6 in a final volume of 25 μl. Fugene6 and DNA were separately diluted in Optimem media (Gibco, 31985070) (1.25 μl final volume each). Fugene6 and DNA solutions were then combined and incubated for 15 min at room temperature. The cell containing solution (22.5 μl) was then added and transfection reactions were mixed and transferred to the assay plate. Following 24 h incubation at 37 °C, wells were rinsed once and 25 μl of HiBit assay buffer (Phenol-red free DMEM F-12 media containing 2% FBS) was added per well. Surface and total receptor expression was measured separately by adding 25 μl per well of the extracellular (Promega, PRN2421) and lytic (Promega, PRN3040) assay solutions, respectively. Plates were incubated for 10 min at room temperature on a shaker and luciferase activity was measured using an EnVision plate reader (Perkin Elmer, Chicago, IL). Expression was normalized as percent of the wild-type receptor signal. Statistical comparisons of expression levels observed with WT versus GPR61 mutants were made by one-way analysis of variance with Dunnett's post hoc test.

**Functional cAMP assays.** A homogeneous Time-Resolved Fluorescence (HTRF) assay (Perkin Elmer, Dynamic2 kit #) was used to assess receptor basal and inverse agonist activity at human GPR61, GPR62, and GPR101. For routine inverse agonist screening frozen CHO TREx human GPR61, GPR62 and GPR101 cells were used. Cells were thawed, resuspended in complete assay media (DMEM F-12, 10% HI FBS, 100 μg/mL Normocin, 700 μg/mL Hygromycin B and 15 μg/mL Blasticidin) containing 1ug/ml doxycycline, plated in white 384 well plates (Corning, 3570) at a density of 750 cells/well (25 μl/ well) and cultured at 37 °C in an incubator overnight. For studies assessing basal and inverse agonist activity at human GPR61 mutants, CHO-K1 cells were transiently transfected using Fugene6, plated in white 384 well plates (7500 cells/well) and grown overnight at 37 °C, as described for

receptor expression studies. The following day, the media was removed and 10 µl of test compound serially diluted in assay buffer (HBSS pH 7.4 containing 20 mM HEPES (Lonza CC-5022), 0.1% BSA (Sigma A7979), 250 µM IBMX (Sigma I7018) was added to the appropriate wells. Compound stock solutions (30 mM) were serially diluted in 100% DMSO and spotted as 120 nL and further diluted with 30ul assay buffer, prior to transfer to the assay plate. Cells and compound were incubated for 60 min in a 37 °C incubator. Cellular cAMP levels were measured following the two-step protocol as per the manufacturer's recommendations. In brief, following incubation 5 µl of cAMP-d2 and 5 ul anti-cAMP cryptate solutions were sequentially added to each well, plates were incubated at room temperature for 1 h and read on an Envision plate reader (PerkinElmer, Waltham, MA, USA). Data were analyzed using the ratio of fluorescence intensity at 620 and 665 nm for each well, extrapolated from the cAMP standard curve to express data as nM cAMP for each well. For compound profiling using stable CHO TREx human GPR61, GPR62 and GPR101 cell line, zero percent effect (ZPE) was defined as nM of cAMP generated from assay buffer/DMSO only. One hundred percent effect (HPE) was defined as nM of cAMP generated in response to 30 uM of a proprietary Pfizer compound with known inverse agonist activity. The % effect values for each compound were plotted by Activity Base using a four-parameter logistic dose response equation, and the concentration required for 50% inhibition ($IC_{50}$) was determined.

For GPR61 WT and mutant profiling using transiently transfected cells, basal and inverse agonist effects were calculated as percent of the wild-type receptor signal measured in the presence of vehicle. The % effect values for each compound were then plotted using GraphPad Prism (v9.5.1) software (version 5.0, San Diego, CA) using a four-parameter logistic dose response equation, and the concentration required for 50% inhibition ($IC_{50}$) was determined. Statistical comparisons of pharmacological parameters observed with WT versus GPR61 mutants were made by one-way analysis of variance with Dunnet's post hoc test.

**Reporting summary**

Further information on research design is available in the Nature Portfolio Reporting Summary linked to this article.

## Data availability

The data that support this study are available from the corresponding author upon request. The cryo-EM maps generated in this study have been deposited in the Electron Microscopy Data Bank under accession codes EMD-41144 (GPR61-G protein complex structure) and EMD-41145 (GPR61 structure with compound 1). The atomic coordinates corresponding to the cryo-EM maps generated in this study have been deposited in the Protein Data Bank under accession codes 8TB0 (GPR61-G protein complex structure) and 8TB7 (GPR61 structure with compound 1). Source data are provided with this paper.

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

## Acknowledgements

The authors gratefully acknowledge K. Sumner, S. Rotstein, D. Tse, and J. Yee for ongoing support of our cryo-EM computing infrastructure.

## Author contributions

J.A.L. performed cryo-EM experiments and model building. J.A.L., J.M.D, and S.H. analyzed the structures. J.A.L., F.R., E.C., and J.M.D. ran AlphaFold 2 predictions and designed constructs. F.R., K.F. and E.C. produced and expressed constructs for cryo-EM. J.M.D. purified proteins for cryo-EM studies. J.M.D., F.R., J.A.L., A.E.V., A.M., and E.C. purified proteins for inactive-state GPR61 construct screening. R.O. and K.A.S. performed and analyzed cAMP activity assays. J.P.F., J.X.K, and E.A.G.H. generated constructs and cell lines for pharmacological studies and performed and analyzed receptor expression studies. Y.Z., J.T., G.L., B.K., R.U., L.Z., and M.M. contributed to design efforts leading to compound 1. B.K. performed compound 1 energy calculations. A.M.D.S., E.F., and D.Z. enabled synthesis of compound 1. Y.Z. supervised the project chemistry. S.H. supervised efforts related to cryo-EM. J.A.L., J.M.D., and S.H. wrote the manuscript, with analysis and input from all authors.

## Competing interests

All authors were employees of Pfizer, Inc. at the time the work described here was performed.
