## [Peer Review File · Nature Communications]

An inverse agonist of orphan receptor GPR61 acts by a G protein-competitive allosteric mechanismREVIEWER COMMENTS

Reviewer #1 (Remarks to the Author):

This manuscript reports the first high resolution structure of GPR61 in active like state with a heterotrimeric G-protein and in inactive states with an inverse agonist ligand. While the active conformation of the receptor might help to understand its constitutive activity and rationalize its selectivity for G α s, the ligand bound structure reveals a novel allosteric pocket. Interestingly, the binding mode of the ligand suggests a new mode of action by remodelling an intracellular pocket used physiologically to block G protein activation. The study used state of the art methodologies and provided a number of novelties including the structural and functional characterisation of the orphan GPR61 receptor and demonstrating a novel site and mechanism of action for blocking its constitutional activity. The manuscript is generally well written, and the results seems to validate its publication in Nature Comm. There are, however, some points that should be commented in a minor revision.

1. The receptor has no sodium site, has no interaction between Glu6.30 and the DRY motif. Therefore, it would be interesting to have any idea on the role of a disulfide bond between Cys2996.47 and Cys3337.45? These features would be expected to weaken interactions known to stabilize the inactive conformation of related receptors, however, this disulfide bond would cause just the opposite effect. Consequently, relatively fewer stabilizing interactions do not obviously effect basal activity.

2. How does the active like structure compare to the state dependent AlphaFold models?

3. In the active structure it seems that extracellular loop (ECL) 2 of GPR61 adopts a lid-like conformation over an orthosteric pocket, with its conformation stabilized by a conserved disulfide bond between Cys115_3.25. Considering, however, that agonist ligands may induce additional conformational changes, and the actual conformation of ECL2 largely occludes the orthosteric pocket that is otherwise plastic, it is more than likely that a potential orthosteric ligand should displace the lid or it is to be contacted by the ligand as in other GPCRs. Consequently, the lid conformation and the small and proportional binding pocket found are probably less relevant for the active structure formed upon binding its endogenous agonist if any exists.

4. Optimization of the GPR61-BRIL construct was based on AlphaFold 2 to retrospectively compare the cryo-EM screening results against their corresponding AlphaFold predictions. Finally, the best construct was further stabilized by BRIL-binding Fab and a hinge-stabilizing nanobody. The resulting structure was compared to that of the state dependent AlphaFold model. Did the authors use AlphaFold2-t or AlphaFold-Multistate for the optimization of the construct? If not, how does the used structure compare

to that? The suggested AlphaFold-informed construct design for the inactive state should probably apply state dependent AF2 models. Can we expect better constructs this way?

5. It would be beneficial to learn more about the identification and optimization of compound 1, the inverse agonist used for the structural studies. Functional data for representative compounds tested during the hit to probe optimization together with SAR data in the closer vicinity of the probe might strengthen the discussion on its novel mechanism of action.

6. The resolution of 3.97 Å was achieved for the receptor-inverse agonist complex, however, the receptor region was poorly resolved. On the other hand, a much better structure with the inverse agonist was obtained (2.9 Å). This indicates that binding the ligand stabilized the complex significantly. Why did the authors keep the artificial GPR61-BRIL complex? This is not biologically relevant.

7. Ligand binding stabilizes the outward conformation of TM6 that resembles in some extent the active state and less the inactive state. Did the authors compare the ligand bound structure systematically and quantitatively to these structures including their AlphaFold Multistate models? What are the specific interactions responsible for the greater similarity of the ligand bound complex to that of the active like conformation?

8. How did the authors select the GPCR targets used for selectivity comparison? The selected set is relevant as secondary pharmacology risk assessment, however, there are more homologous GPCRs e.g., GPR62, HRH2, ADA1D, 5HT6R, ADA1B, 5HT1E and ADA1A.

Reviewer #2 (Remarks to the Author):

Lees and Dias et al. present the first CryoEM structures of GPR61 with and without an inverse agonist. Moreover, this inverse agonist is shown to bind at a novel intracellular binding site overlapping with Gas unlike recently reported ICL binding sites for GPCRs. GPR61 is an orphan receptor, and seems to be an interesting drug target for e.g. metabolic disorders. Hence, this study in my opinion aids drug discovery for this target. However, some amendments are needed for publication in Nature communications.

Major points:

- For CryoEM studies constructs have been developed and used. However, the authors do not show that these constructs behave similar to WT (as done in many/most structure elucidation papers). I realize

that it is impossible to do this functionally, but the authors should think of other ways to proof that the constructs behave WT-like, e.g. cellular localization receptor expression. This is a crucial control that should be done.

- In general the structure of the paper should be revised; some information provided in the results section (e.g. page 4; line 68-74) fits better in introduction. Moreover, the discussion is more of an extensive summary (i.e. no citations are used at all), so either results and discussion should be combined (if journal allows) or discussion should be moved from results section to create a real discussion section.

- I believe more information can/should be obtained from the apo-structure than is currently presented. For example, a statement on ECL2 is made that it adopts a lid-like conformation much like other class A GPCRs, and that unlike many other receptors this loop occludes the orthosteric binding pocket. However, no experimental (comparative modeling) data is provided for this. Moreover, could this occlusion of the orthosteric pocket cause ECL2 to activate the receptor? Since the endogenous agonist is unknown, would this explain its constitutive activity?

- Provide some more background on the Pfizer internal compound as it is used as a control, e.g. chemical structure, expected binding site

- Compound 1 is stated to be selective. But when expecting the assays performed (extended data table 1), it is striking to see that: 1) no other Gs-coupled receptors (or readout) is being done, and 2) the GPCRs with reported ICL binding sites are missing. This needs to be repaired.

- It is noted that the increased receptor expression in presence of compound 1 substantiates its inverse agonistic mode of action. However, it could also be functioning as a pharmacochaperone. Is a similar observation done with the Pfizer compound?

- Some points on the in vitro data presented in Figure 2; 1) DRCs for both tagged and untagged receptors, and for both Pfizer-compound and compound 1 (in Figure and Table). 2) In stead of showing bar graphs of basal cAMP levels (3f), provide the data in the table (3e). 3) Move comparison of tagged- and untagged receptors to supplemental. 4) WT basal cAMP activity does not seem to have error bars; how is significance determined? 5) WT-vehicle is set at 100%, what is used for 0%? 6) The general scheme should be removed here and merged with Figure 5 (as this also entails a cartoon)

- Most points above on Figure 2 also account for Figure 3. Moreover, the authors should show the mutant data for compound 1.

Minor points:

- The authors only mention 5-(Nonyloxy)Tryptamine as a Low-Affinity Inverse Agonist (page 3; line 45), but ethanolamine plasmalogens (EPIs) have also been mentioned as ligands in literature.

- A case is made that apo-structures are difficult to obtain (page 3; line 47), which is supported by a statement of the general paucity of apo-GPCR structures in the PDB. It would be good to be more specific here, what is the current ratio of liganded vs. unliganded structures? (focus on CryoEM, as this seems to be less of a problem)

- GPR61 is thought to be a constitutively active receptor. Has it been excluded that the ligand is not present in serum or culture medium? Related to this, the authors should be more careful to state compound 1 (or the Pfizer reference) as an inverse agonist. It could be an allosteric antagonist.
- Please make clear that after the sentence “as other key sequence... the inactive state”, some example are given to substantiate this statement.
- It is shown that the N345A mutant causes a decreased level of constitutive activity (page 1-; line 220) and that this is caused by less membrane expression. However, in case of the ligand (see above) decreasing the receptor’s constative activity is used as an explanation for finding an increased expression level. This is contradictory, please clarify.
- What is meant with the “bioactive conformation” of compound 1? (page 11; line 227)
- On page 11 compounds 1 potency and selectivity are being explained. For the residues interacting with the sulfonamide moiety it is clearly stated that these are key to potency but less to selectivity as they are more conserved. While for the Asparagine only a case is made for the compounds potency and stated that is not conserved. Is this then crucial for compound 1’s selectivity? An alignment for the GPCRs tested in the selectivity assays for these residues would be valuable.
- On page 13; line 292; *in silico* should written in italic.
- There seems to be an erroneous full stop on page 13; line 294.
- In some cases IC₅₀ is written as IC50, i.e. 50 should be in subscript.
- Page 27; line 501: A₂AR should be A2AR (subscript)
- Page 40; line 728-732; some data seems to be missing here. Moreover the Y-axis of Figure 2d and 3e indicate that vehicle is set at 100% (while here it is state that vehicle is set at ZPE). The term ZPE and HPE are not used anywhere else in the manuscript.

Reviewer #3 (Remarks to the Author):

This manuscript presents cryo-EM structures of both inactive and active states of GPR61, providing valuable molecular insights into the modulation of this receptor. The authors first determined the structure of GPR61 complexed with an engineered Gs heterotrimer, shedding light on the mechanism underlying the high constitutive activity of GPR61. Subsequently, they identified a novel inverse agonist, Compound 1, and elucidated the structures of inactive GPR61 alone and in complex with Compound 1. To achieve the cryo-EM structures of inactive GPR61, the authors utilized AlphaFold2 to guide their protein engineering strategy and designed a GPR61-BRIL fusion construct with continuous helicity at the junctions between TM5/6 and the BRIL helices. Notably, the structures of apo- and Compound 1-bound GPR61 revealed a ligand-induced allosteric site for Compound 1, which has not been observed in other GPCRs. Based on the structures and mutagenesis studies, the authors further discussed the mechanism by which Compound 1 inhibits GPR61 signaling. Overall, this is an outstanding structural study on an

orphan GPCR, with well-supported conclusions drawn from both structural and functional data. The cryo-EM structural models are well supported by their maps, especially for Compound 1. The figures presented are clear and informative, further enhancing the manuscript's impact.

I only have some minor comments and issues:

1. The 'ionic lock' hypothesis was originally proposed based on structural studies of rhodopsin, which is required to remain completely inactive in the absence of light. The 'ionic lock' mechanism is thought to help maintain this inactive state by creating a stable interaction between the polar residues D3.50 and E6.30 in transmembrane helices 3 and 6, respectively. However, subsequent research has revealed that many Class A GPCRs do not possess this 'ionic lock' between D3.50 and E6.30 or other polar residues in TM6, indicating that it is not a universal mechanism for maintaining receptor inactivity in the absence of agonists. Therefore, lacking such 'ionic lock' may not be so special for GPR61 to explain its high basal activity.

2. On the other hand, the distinctive disulfide bond formed between C6.47 and C7.45, located just above W6.48, is particularly intriguing. The microswitch residues W6.48 and F6.44 are typically involved in linking agonist binding to the outward displacement of TM6 in Class A GPCRs through conformational changes. It is possible that the C6.47 and C7.45 disulfide bond could somehow influence the conformation of W6.48, thereby facilitating the receptor's transition to the active state. Further mutagenesis studies targeting these two Cys residues or computational simulations could yield important insights into this potential mechanism.

3. Is it possible that the BRIL insertion somehow affects the conformation of TM6 causing the binding site of Compound 1 to be closed in the apo structure? Does the AlphaFold2 predicted structure have the allosteric site?

4. Residues in the Disallowed Ramachandran Plot areas need to be adjusted for structures below 4Å resolution. Also, the clash score of the GPR61-G protein complex structure is too high.

5. The authors used CryoSparc to process their cryo-EM data. However, one may wonder if there was a specific reason for not employing heterogeneous refinement to identify more homogeneous particle subsets?

Reviewer #1

This manuscript reports the first high resolution structure of GPR61 in active like state with a heterotrimeric G-protein and in inactive states with an inverse agonist ligand. While the active conformation of the receptor might help to understand its constitutive activity and rationalize its selectivity for Gas, the ligand bound structure reveals a novel allosteric pocket. Interestingly, the binding mode of the ligand suggests a new mode of action by remodelling an intracellular pocket used physiologically to block G protein activation. The study used state of the art methodologies and provided a number of novelties including the structural and functional characterisation of the orphan GPR61 receptor and demonstrating a novel site and mechanism of action for blocking its constitutional activity. The manuscript is generally well written, and the results seems to validate its publication in Nature Comm. There are, however, some points that should be commented in a minor revision.

We thank the reviewer for taking the time to examine our manuscript and for their kind comments regarding the significance of our work. We have attempted to address each of the reviewer's concerns point by point below.

1. The receptor has no sodium site, has no interaction between Glu6.30 and the DRY motif. Therefore, it would be interesting to have any idea on the role of a disulfide bond between Cys2996.47 and Cys3337.45? These features would be expected to weaken interactions known to stabilize the inactive conformation of related receptors, however, this disulfide bond would cause just the opposite effect. Consequently, relatively fewer stabilizing interactions do not obviously effect basal activity.

We recognize the reviewer's concern regarding the role of the novel disulfide between TM6 and TM7 of GPR61. The proximity of this disulfide to the key tryptophan rotamer toggle switch (one of the participating cysteines, Cys299, is the residue preceding the switch, Trp300^{6.48}) suggested that it may have a role in activation. To provide further insight and experimentally evaluate this interpretation, however, we have examined the basal cAMP activity and cell surface expression of a mutant receptor (C299S) that abolishes this disulfide bond. We observed that the mutant construct exhibits a similar level of surface expression-normalized basal activity to the wild-type receptor. This suggests that breakage of this disulfide has little impact on the basal activity of the receptor and thus it may not make meaningful contributions to GPR61's constitutive activation. We have amended the text to reflect this new observation, as follows:

Lines 131-134 (revised no markup version):

“To probe the function of this disulfide, we generated a GPR61 mutant (C299S) unable to form a disulfide bond, but found that it exhibits a similar level of basal activity to the wild-type receptor (Extended Data Figure 1), suggesting that it may not play a significant role in basal activation.”

2. How does the active like structure compare to the state dependent AlphaFold models?

Our active-state cryo-EM structure, when aligned to the active-state AlphaFold model, exhibits an RMSD of 2.44 Å. The bulk of the difference between the models is due to an outward displacement

of the intracellular portion of TM6 and an inward displacement of the intracellular portions of TM7 and TM1 in the cryo-EM structure relative to the AlphaFold model. Each of these changes is generally associated with most GPCRs' transition from the inactive to active state to accommodate binding of the G α protein. Indeed, the position of TM6 in the active state AlphaFold model would create clashes with the C-terminal helix of G α s in our structure. We have made the following modifications to the text:

Lines 325-327 (revised no markup version):

“Interestingly, however, the position of TM6 in the AlphaFold MultiState prediction for active GPR61 shows relatively poor agreement with our active-state cryo-EM structure, being positioned inward slightly, where it would clash with the position of G α s.”

3. In the active structure it seems that extracellular loop (ECL) 2 of GPR61 adopts a lid-like conformation over an orthosteric pocket, with its conformation stabilized by a conserved disulfide bond between Cys115_3.25. Considering, however, that agonist ligands may induce additional conformational changes, and the actual conformation of ECL2 largely occludes the orthosteric pocket that is otherwise plastic, it is more than likely that a potential orthosteric ligand should displace the lid or it is to be contacted by the ligand as in other GPCRs. Consequently, the lid conformation and the small and proportional binding pocket found are probably less relevant for the active structure formed upon binding its endogenous agonist if any exists.

We acknowledge the reviewer's observation that we might expect significant changes to the arrangement of ECL2 upon agonist binding, which may include its displacement from the orthosteric site. Alternatively, ECL2 could remain in place and form a part of an expanded binding site opened by structural plasticity in the N-terminal region of TM1, which is poorly ordered in our structure. While we have commented on the relatively small volume of the orthosteric pocket in the manuscript, on the basis of our data we cannot yet speak to how the shape of the orthosteric pocket may change in the presence of an agonist. To reduce the chance of misleading potential readers, we have added a clarifying statement to the text to make this more explicit, as duplicated below.

Lines 86-90 (revised no markup version):

“Diffuse density for the extracellular portion of TM1 suggests the pocket is plastic, and displacement of ECL2 by a ligand, as for instance in the case of rhodopsin^{21,22}, could expose and/or frame a larger and deeper pocket. Thus, we cannot predict how the conformation of ECL2 might differ from the observed structure in the presence of a potential agonist.”

4. Optimization of the GPR61-BRIL construct was based on AlphaFold 2 to retrospectively compare the cryo-EM screening results against their corresponding AlphaFold predictions. Finally, the best construct was further stabilized by BRIL-binding Fab and a hinge-stabilizing nanobody. The resulting structure was compared to that of the state dependent AlphaFold model. Did the authors use AlphaFold2-t or AlphaFold-Multistate for the optimization of the construct? If not, how does the used structure compare to that? The

suggested AlphaFold-informed construct design for the inactive state should probably apply state dependent AF2 models. Can we expect better constructs this way?

At the time that we were designing the constructs for this study, AlphaFold Multistate was not yet available to us, so our constructs were designed using AlphaFold2. When the AlphaFold2 prediction for our final BRIL insertion construct is compared to the prediction for wild-type GPR61, it exhibits significant deformations of TM5 and TM6 induced by the BRIL. While wild-type TM6 from the AlphaFold2 prediction adopts an intermediate position between the two extremes predicted by AlphaFold Multistate, the positions of both TM5 and TM6 of the BRIL insertion construct's AlphaFold2 prediction agree strikingly well with the Multistate prediction for the wild-type inactive form. This would seem to argue that even the standard AlphaFold2 predictions can achieve similar results to AlphaFold Multistate when screening such constructs, though this example clearly does not represent an exhaustive analysis. It should be emphasized that we view AlphaFold2 as a tool to triage potential constructs, identifying a smaller subset for experimental screening. In its current form(s), it is unlikely to pinpoint a single best construct. We have added text addressing this concern to the discussion section:

Lines 318-324 (revised no markup version):

“While AlphaFold 2 was used to screen the BRIL fusion constructs used for the inactive-state structure, a subsequent comparison of the AlphaFold 2 GPR61_{IA} prediction to the inactive-state AlphaFold MultiState prediction for wild-type GPR61 revealed that they are in striking agreement, while the AlphaFold 2 prediction for wild-type GPR61 adopts a conformation intermediate between the active and inactive states. This suggests that AlphaFold2 may be a reliable tool for screening such fusions, but it is possible that improved predictions could be obtained using AlphaFold MultiState.”

5. It would be beneficial to learn more about the identification and optimization of compound 1, the inverse agonist used for the structural studies. Functional data for representative compounds tested during the hit to probe optimization together with SAR data in the closer vicinity of the probe might strengthen the discussion on its novel mechanism of action.

We are sympathetic to the reviewer's opinion that discussion of the process underlying the design of compound 1 would provide useful insights relevant to its mechanism of action. While the details underlying the development of compound 1 are beyond the scope of this manuscript, a detailed and thorough analysis of the optimization of this compound, including SAR, will be presented in a separate medicinal chemistry manuscript currently in preparation.

6. The resolution of 3.97 Å was achieved for the receptor-inverse agonist complex, however, the receptor region was poorly resolved. On the other hand, a much better structure with the inverse agonist was obtained (2.9 Å). This indicates that binding the ligand stabilized the complex significantly. Why did the authors keep the artificial GPR61-BRIL complex? This is not biologically relevant.

The artificial GPR61-BRIL complex was a necessary tool to obtain the structure, despite its lack of biological relevance. While the presence of compound 1 does indeed stabilize the structure of GPR61, it cannot take the place of the critical particle alignment fiducial required for cryo-EM that is provided by the G protein complex in the activated form. Since the presence of compound 1 precludes the assembly of the G protein complex, some other sort of fiducial feature external to the micelle is necessary to allow particle alignment during cryo-EM data processing. This is particularly true for class A GPCRs like GPR61, which have no soluble domains. In our study, an alternative fiducial was provided by the BRIL insertion in combination with the BRIL Fab and hinge-binding nanobody, based on the similar strategy used by Tsutsumi et al., as described in the main text. While other engineered tool constructs have been used for this purpose (for example, Nb6, as described in Robertson et al., “Structure determination of inactive-state GPCRs with a universal nanobody”, Nat Struct Mol Biol 2022), they each serve the same function of providing a cryo-EM alignment fiducial.

7. Ligand binding stabilizes the outward conformation of TM6 that resembles in some extent the active state and less the inactive state. Did the authors compare the ligand bound structure systematically and quantitatively to these structures including their AlphaFold Multistate models? What are the specific interactions responsible for the greater similarity of the ligand bound complex to that of the active like conformation?

Our reference to the compound 1-bound structure as adopting a more active-like conformation than the canonical inactive state refers primarily to the outward position of the intracellular portion of TM6, similar to its position that would ordinarily allow the receptor to accommodate the binding of the G α s protein. This similarity, however, is limited and driven by the interactions made by compound 1 with the protein, as detailed in the manuscript. Furthermore, there are key differences in this pocket between the active state and the compound-bound state, as described by the text associated with figure 4a. This is discussed in the text as “remodeling” of the G α s-binding pocket. To avoid any overly strong implication of deeper structural similarity to the active state, we have made the following modifications to the text:

Lines 205-209 (revised no markup version):

“The formation of this induced binding pocket is enabled by a counter-intuitive conformational change in which the intracellular half of TM6 is forced outward relative to its position in the apo structure, more closely resembling the active than the inactive form of the receptor (Fig. 3c), though the detailed interactions differ significantly from the active state.”

Lines 265-267 (revised no markup version):

“As discussed above, when bound to compound 1, TM6 of the receptor adopts a conformation resembling that of the active receptor, but that nonetheless precludes the binding of G α s necessary for activation of downstream signaling.”

8. How did the authors select the GPCR targets used for selectivity comparison? The selected set is relevant as secondary pharmacology risk assessment, however, there are more homologous GPCRs e.g., GPR62, HRH2, ADA1D, 5HT6R, ADA1B, 5HT1E and ADA1A.

We concede the reviewer's point that the selectivity panel reported here in Extended Data Table 1 is geared more toward secondary pharmacological risk assessment. While it was not possible to perform selectivity analysis with all of these receptors, we have performed a multiple sequence alignment (See Extended Data Fig. 7) that includes all of the receptors included in our original selectivity panel, as well as the receptors you cite here. Residues lining the compound 1-binding pocket, including the key interacting residues detailed in figure 3, are highlighted. We selected two close homologs that signal through G α s (GPR62 and GPR101) to measure their response to compound 1 (ADA1A was already included in the pharmacological risk assessment panel) for comparison to GPR61. The data are reported in Extended Data Table 2 and confirm that compound 1 has roughly 1000-fold lower potency against these receptors than against GPR61, in line with observations of the other receptors in the original selectivity panel. It's worth noting that the residue identified as critical for selectivity of compound 1, Asn137, is found here only in GPR61. The results of these analyses remain consistent with the hypothesis that Compound 1 is selective for GPR61. We have modified the manuscript text to note this, as follows:

Lines 247-249 (revised no markup version):

“Compound 1 was found to be selective for GPR61 in a panel of common off-target GPCRs, as well as against related receptors GPR62 and GPR101 (Extended Data Table 1, 2, Extended Data Figure 7).”

Reviewer #2

Lees and Dias et al. present the first CryoEM structures of GPR61 with and without an inverse agonist. Moreover, this inverse agonist is shown to bind at a novel intracellular binding site overlapping with Gas unlike recently reported ICL binding sites for GPCRs. GPR61 is an orphan receptor, and seems to be an interesting drug target for e.g. metabolic disorders. Hence, this study in my opinion aids drug discovery for this target. However, some amendments are needed for publication in Nature communications.

We thank the reviewer for their time in examining our manuscript and for their recognition of the value of our findings. We have attempted to address each of the reviewer's concerns below.

Major points:

- For CryoEM studies constructs have been developed and used. However, the authors do not show that these constructs behave similar to WT (as done in many/most structure elucidation papers). I realize that it is impossible to do this functionally, but the authors should think of other ways to proof that the constructs behave WT-like, e.g. cellular localization receptor expression. This is a crucial control that should be done.

As the reviewer correctly notes, functional analysis of the GPR61 BRIL fusion is unfortunately precluded by the nature of our modifications to the construct. However, in response to the reviewer's request, we have made a comparison of the cell surface expression of the inverse agonist cryo-EM construct against the equivalent construct lacking the BRIL insertion (wild-type receptor sequence). The results indicate that both constructs exhibit high levels of cell surface expression

in insect cells, with the BRIL fusion in fact being expressed at the cell surface at a much higher level than the wild-type receptor, possibly due to structural stabilization caused by the BRIL fusion. We have added these data to the manuscript's supplemental materials (Extended Data Figure 3), added corresponding text to the methods, and referenced the analysis with the following text in the main body of the manuscript:

Lines 183-186 (revised no markup version):

“While functional analysis of this construct was not possible with the BRIL fusion, a comparison of the optimized construct's expression at the cell surface against that of the wild-type receptor indicated that it is properly trafficked to the cell surface and thus well-folded (Extended Data Figure 3b,c).”

- In general the structure of the paper should be revised; some information provided in the results section (e.g. page 4; line 68-74) fits better in introduction. Moreover, the discussion is more of an extensive summary (i.e. no citations are used at all), so either results and discussion should be combined (if journal allows) or discussion should be moved from results section to create a real discussion section.

We acknowledge the reviewer's observation and have attempted to strike a better balance between the results and discussion sections by moving some more discussion-like sections of the results text (including the specific text referenced by the reviewer) to the discussion. We have also included some additional discussion of the AlphaFold strategy and rearranged the text of the discussion section to improve its flow. While the complete suite of changes cannot be conveniently duplicated here, the revised text includes tracked changes.

- I believe more information can/should be obtained from the apo-structure than is currently presented. For example, a statement on ECL2 is made that it adopts a lid-like conformation much like other class A GPCRs, and that unlike many other receptors this loop occludes the orthosteric binding pocket. However, no experimental (comparative modeling) data is provided for this. Moreover, could this occlusion of the orthosteric pocket cause ECL2 to activate the receptor? Since the endogenous agonist is unknown, would this explain its constitutive activity?

Comparative modeling of ECL2 against related (liganded) receptors indicates that the trajectory of the latter portion of GPR61 ECL2 does not differ dramatically from most others. Unlike these related receptors, however, the orthosteric site beneath the loop is considerably smaller, due to the presence of several bulky residues lining the pocket, including Phe212 and His307. It does differ markedly, however, from receptors for which the so-called ECL2 “agonist-like motif” has been identified, including GPR52 (Lin et al., “Structural basis of ligand recognition and self-activation of orphan GPR52”, Nature 2020) and GPR21 (Wong et al., “Cryo-EM structure of orphan G protein-coupled receptor GPR21”, MedComm 2023; Lin et al., “Cryo-EM structures of orphan GPR21 signaling complexes”, Nat Commun 2023). The agonist-like motifs of these receptors share similar trajectories and structures, but are dissimilar to the GPR61 ECL2 and protrude much deeper into the orthosteric pocket. To experimentally address a potential role of ECL2 in basal receptor activation, we have tested the basal cAMP activity of several mutant versions of the receptor, modeled on a similar set of mutations used to probe the role of the agonist-like motif of

GPR52 (Lin et al., 2022). We mutated key residues of GPR61 ECL2 to disrupt the conserved disulfide (C195A) or to replace key residues that dip into the orthosteric pocket (W199A, Q198A). An additional mutant replaced residues 194-201 of GPR61 ECL2 with a flexible Gly-Ser linker, which also disrupts the conserved disulfide. The results, presented in Extended Data Figure 7, indicate that disruption of the conserved disulfide by the C195A mutation or by replacement of a portion of the loop sequence with a Gly-Ser linker appears to decrease the receptor's constitutive activity by about 50% when corrected for cell surface expression. Point mutation of residues that dip into the orthosteric pocket (Q198A, W199A) appears to have only a minimal effect on the basal activity of the receptor (although the Q198A mutation yields slightly increased basal activity). Overall, these observations suggest that while the overall conformation of ECL2 may mildly affect the level of GPR61 constitutive activity, even the most drastic of our mutations did not abolish constitutive signaling. In contrast, the mutations made to the agonist-like motif of GPR52, GPR21, and GPR12 universally decreased their constitutive activity by more than 80%. Thus, we conclude that ECL2 of GPR61 likely does not contribute significantly to the constitutive activity of GPR61. We have made the following amendments to the manuscript text to reflect our observations:

Lines 135-143 (revised no markup version):

“An agonist-like motif (ALM) found in ECL2 has recently been identified in several receptors, including GPR52, GPR21, and GPR12³⁰⁻³³ as a stimulator of constitutive receptor activity. To probe the potential role of ECL2 in GPR61 activation, we examined the basal activity of GPR61 with point mutations in ECL2 designed to weaken its interactions with the orthosteric site (Q198A, W199A) and/or abolish the conserved disulfide (Δ 195-201(GGSGGSGG), C195A) (Extended Data Figure 1). While mutants that abolished the disulfide modestly decreased constitutive activity, none of the mutants caused the catastrophic loss of activity observed for similar mutants of GPR52, GPR21, and GPR12, suggesting that ECL2 is not a major contributor to GPR61's constitutive activity.”

- Provide some more background on the Pfizer internal compound as it is used as a control, e.g. chemical structure, expected binding site.

We interpret the reviewer's comment to refer to one of two things, which we address separately.:

-The compound used as a control to define one hundred percent effect (HPE) in our assays is a non-selective inhibitor that acts by an unknown mechanism. While we have not revealed its structure here, we have measured its activity against the basal cAMP activity of GPR61, GPR62, and GPR101. The compound is similarly potent (i.e. non-selective) against all three receptors. These data are included in Extended Data Table 2.

-Alternatively, if the reviewer's comment instead refers to the PF compound originally named in the axis label of figure 3e, this is due to our original mislabeling of the axis. The data represented in figure 3e show the effects of compound 1. The labeling error has been corrected in the revised manuscript.

- Compound 1 is stated to be selective. But when expecting the assays performed (extended data table 1), it is striking to see that: 1) no other Gs-coupled receptors (or readout) is being

done, and 2) the GPCRs with reported ICL binding sites are missing. This needs to be repaired.

The reviewer's point is well-taken. To address the reviewer's concern regarding receptors that signal through G α s, as does GPR61, we have additionally measured compound 1's inverse agonist activity against GPR62 and GPR101, both of which signal through G α s. These data, reported in Extended Data Table 2, show compound 1 to be about 1000-fold more potent against GPR61 than these receptors. While an exhaustive panel of receptors known to harbor intracellular inverse agonist binding pockets was not possible, they are represented here by the β 2-adrenergic receptor, for which results were originally reported in Extended Data Table 1. We have included a multiple sequence alignment in Extended Data Figure 7 to allow comparison of key residues lining the compound binding site in these receptors for convenience. It bears repeating that the binding site occupied by compound 1 is novel, and the participating residues of GPR61 do not significantly overlap with those of any of the other known GPCRs' intracellular allosteric pockets. We have made mention of the additional results in the text, as follows:

Lines 247-248 (revised no markup version):

“Compound 1 was found to be selective for GPR61 in a panel of common off-target GPCRs, as well as against related receptors GPR62 and GPR101 (Extended Data Table 1, 2; Extended Data Figure 7).”

- It is noted that the increased receptor expression in presence of compound 1 substantiates its inverse agonistic mode of action. However, it could also be functioning as a pharmacochaperone. Is a similar observation done with the Pfizer compound?

We cannot exclude the possibility that compound 1 acts as a pharmacochaperone to increase cell surface expression of GPR61. Nonetheless, its definition as an inverse agonist is established on the basis of its structure-validated molecular mechanism and its impact on basal cAMP activity. The observation of increased cell surface expression of GPR61 in the presence of compound 1 could have multiple explanations. We have proposed that it may be a downstream compensatory effect arising from reduced signaling by the receptor, but this hypothesis itself is not a critical element of the argument to substantiate Compound 1's inverse agonism, and other hypotheses, such as pharmacochaperone activity, cannot be excluded by the data. We have modified the text to make this clearer.

Lines 149-154 (revised no markup version):

“Consistent with a role in suppressing constitutive receptor activity, compound 1 caused increased cell surface expression of a HiBit-tagged form of wild-type GPR61 (Fig. 2b, c). We attribute this to compensatory overexpression in response its inhibition, though we cannot exclude the possibility that a different cause, such as pharmacochaperone activity, might account for increased surface expression of the receptor.”

- Some points on the in vitro data presented in Figure 2; 1) DRCs for both tagged and untagged receptors, and for both Pfizer-compound and compound 1 (in Figure and Table). 2) In stead of showing bar graphs of basal cAMP levels (3f), provide the data in the table (3e). 3) Move comparison of tagged- and untagged receptors to supplemental. 4) WT basal

cAMP activity does not seem to have error bars; how is significance determined? 5) WT-vehicle is set at 100%, what is used for 0%? 6) The general scheme should be removed here and merged with Figure 5 (as this also entails a cartoon)

Regarding the reviewer's comments here, we will respond point by point:

1) Given that the table in figure 3e includes both tagged and untagged IC₅₀ values, we have added the dose-response curves for the HiBit-tagged construct to the figure panel. The compound the reviewer refers to as the Pfizer compound has been explained elsewhere and is the same as compound 1. It was mislabeled in the original version of the manuscript and we have corrected the label.

2) The data in figure 3h (now 3f) derive from those in former figure panels 3f and 3g and thus are the most important to present in the figure. Thus, we have moved the data in figures 3f and g to the supplement (Extended Data Figure 5). Because the types of data reported in 3e and 3h (now 3f) describe different parameters (compound 1 IC₅₀ values vs. basal activity measurements), we respectfully request to keep them separated from one another.

3) Since figure 3h (now 3f) provides the data most relevant to evaluating the basal activity of the mutants, we have left it in the main figure, but have moved the supporting data in 3f and g to supplement, as explained in point 2.

4) Data are expressed as a percentage of the wild-type value to minimize the contribution of day-to-day variation to the noise in the data. Statistical comparisons of pharmacological parameters observed with WT versus GPR61 mutants were made by one-way analysis of variance with Dunnett's post hoc test, which assumes that the values are normally distributed and that there is an even distribution of variability in the groups. This type of analysis has been used in similar GPCR pharmacology studies:

- Koole et al. Mol Pharmacol. 2011. 80(3):486-97
(<https://pubmed.ncbi.nlm.nih.gov/21616920/>)
- Fortin et al. Mol Pharmacol. 2010. 78(5):837-45
(<https://pubmed.ncbi.nlm.nih.gov/20702761/>)

5) In the current analysis, we have not defined 0%. We could set 0% as the residual basal cAMP activity observed in the presence of 10 μM compound 1 at WT or a mutant. However, since such modification would not change our conclusions, we prefer a simpler normalization as % WT vehicle.

6) We respectfully request to maintain figure panel 2e in its current position, as it is provided for convenient comparison to the data shown in 2d. We have clearly indicated in the figure legend that this does not represent experimental data, and we believe that moving it to figure 5 will significantly dilute its impact.

- Most points above on Figure 2 also account for Figure 3. Moreover, the authors should show the mutant data for compound 1.

We believe the reviewer's comment refers to an instance of mislabeling on figure 3e on the x-axis of the dose-response curves. The compound used in these experiments was indeed compound 1 and we have corrected the labeling of figure 3e to reflect this.

Minor points:

- The authors only mention 5-(Nonyloxy)Tryptamine as a Low-Affinity Inverse Agonist (page 3; line 45), but ethanolamine plasmalogens (EPIs) have also been mentioned as ligands in literature.

We thank the reviewer for pointing this out. We recognize that EPIs have been proposed as potential agonists of GPR61 and regret our oversight in not including them in the text. Accordingly, we have added a statement mentioning EPIs, paired with an appropriate reference, to the text, as duplicated below:

Lines 45-46 (revised no markup version):

“Ethanolamine plasmalogens have been suggested as endogenous GPR61 ligands¹⁰, but further study is needed to better understand this activity.”

- A case is made that apo-structures are difficult to obtain (page 3; line 47), which is supported by a statement of the general paucity of apo-GPCR structures in the PDB. It would be good to be more specific here, what is the current ratio of liganded vs. unliganded structures? (focus on CryoEM, as this seems to be less of a problem)

In the main text, we have provided counts of liganded and apo GPCR structures solved by cryo-EM as reported in the GPCR database on April 20, 2023: of 524 total cryo-EM structures (130 receptors), 55 are apo (37 receptors). The addition to the text is duplicated below:

Lines 52-54 (revised no markup version):

“This is reflected in a relative paucity of unliganded GPCR structures in the Protein Data Bank (out of 524 GPCR structures solved by cryo-EM representing 130 receptors, as reported by the GPCR database¹³, only 55 structures of 37 receptors are without a ligand).”

- GPR61 is thought to be a constitutively active receptor. Has it been excluded that the ligand is not present in serum or culture medium? Related to this, the authors should be more careful to state compound 1 (or the Pfizer reference) as an inverse agonist. It could be an allosteric antagonist.

We acknowledge the reviewer's point that an as-yet-unidentified agonist could be present in the environment in which the expressing cells were grown, which we cannot categorically exclude. Because compound 1 abolishes GPR61 signaling activity through direct competition with binding of the G protein, based on our cryo-EM structures, its mechanism would necessarily make it an inverse agonist, so long as GPR61 is recognized to exhibit constitutive activity. The field's current consensus is that GPR61 is a constitutively active receptor, and undertaking a rigorous study of the growth medium to confirm or contradict that consensus is beyond the scope of this study.

Therefore, in the absence of contradictory data, we refer to compound 1 as an inverse agonist in recognition of the field's current consensus that GPR61 exhibits constitutive activity. Furthermore, it's worth noting that our manuscript reports the structure of GPR61-G protein complex in its active state, but we find no evidence of an agonist occupying the receptor's orthosteric site in that structure. We have, however, added a statement to the manuscript that acknowledges the possibility of an unidentified agonist being present, as follows:

Lines 302-305 (revised no markup version):

“Although constitutive activation of GPR61 was suggested to involve the receptor's N terminus, the structural basis of its constitutive activation is unknown, and the possibility of a yet-to-be-identified agonist present under expression conditions has not been categorically excluded.”

- Please make clear that after the sentence “as other key sequence... the inactive state”, some example are given to substantiate this statement.

This sentence in the main text is intended to introduce the structural features we observe in the GPR61 structure that are indicative of a less stable inactive conformation, such as the absence of the ionic lock and sodium binding site. To avoid confusion, we have altered the wording of this sentence to clarify that it precedes the listing of these characteristics of GPR61:

Lines 113-116 (revised no markup version):

“The bias of GPR61 toward constitutive activation may not rely solely on its N-terminal peptide, however, as the following key sequence and structural features could be consistent with partial destabilization of the inactive state.”

- It is shown that the N345A mutant causes a decreased level of constitutive activity (page 1- ; line 220) and that this is caused by less membrane expression. However, in case of the ligand (see above) decreasing the receptor's constative activity is used as an explanation for finding an increased expression level. This is contradictory, please clarify.

Because the direction of causation is opposite in these two situations, we do not see these two statements as contradictory. To clarify, the N345A mutant exhibits reduced cell surface expression, and our statement was intended to be a cautious interpretation of the reduced activity of this mutant and was proposed as a possible membrane trafficking defect caused by the mutation. Nonetheless, it would be expected that its reduced cell surface expression would give rise to reduced signaling on the basis of its lower availability, and indeed its basal activity is similar to wild-type when normalized to cell surface expression. In contrast, in the presence of inverse agonist compound 1, signaling through the G protein is impaired, which may give rise to a compensatory increase in receptor expression. Because the compound can act on any newly synthesized receptor, as well, the increased expression may not be accompanied by a recovery of activity. To address this concern, we have modified the relevant sections of the manuscript to be clearer:

Lines 149-154 (revised no markup version):

“Consistent with a role in suppressing constitutive receptor activity, compound 1 caused increased cell surface expression of a HiBit-tagged form of wild-type GPR61 (Fig. 2b, c). We attribute this

to compensatory overexpression in response its inhibition, though we cannot exclude the possibility that a different cause, such as pharmacochaperone activity, might account for increased surface expression of the receptor.”

- What is meant with the “bioactive conformation” of compound 1? (page 11; line 227)

In this statement, we use the phrase “bioactive conformation” to refer to the conformation of compound 1 when bound to GPR61, i.e. when it is exerting its inverse agonist effect. We have altered the wording in the manuscript to be clearer, as follows:

Lines 213-215 (revised no markup version):

“The bound conformation of compound 1 wraps around the side chain of Val288^{6,36}, forming extensive stabilizing Van der Waals’ contacts (Fig. 3d).”

Lines 236-237 (revised no markup version):

“The similarity of compound 1’s GPR61-bound conformation to its global energetic minimum conformation likely also contributes to its potency.”

- On page 11 compounds 1 potency and selectivity are being explained. For the residues interacting with the sulfonamide moiety it is clearly stated that these are key to potency but less to selectivity as they are more conserved. While for the Asparagine only a case is made for the compounds potency and stated that is not conserved. Is this then crucial for compound 1’s selectivity? An alignment for the GPCRs tested in the selectivity assays for these residues would be valuable.

Asn137^{3,47} is indeed a key residue responsible for the selectivity of compound 1, given its rarity among GPCRs and the clear contribution it makes to the potency of compound 1. To make this clearer in the text, we have modified the text as follows:

Lines 253-259 (revised no markup version):

“As a key contributor to the compound’s potency, mutation of Asn137 would be expected to exact a large energetic penalty, reducing the compound binding affinity considerably. Consistent with this hypothesis, mutation of Asn137^{3,47} to Ala in GPR61 reduced the potency of compound 1 about 100-fold relative to wild-type in the cAMP assay (Fig. 3e), making it a key contributor to compound 1’s selectivity. In contrast, the residues interacting with the sulfonamide moiety lend potency, but are much more highly conserved and thus do not contribute significantly to selectivity.”

We have also added a sequence alignment, as requested by the reviewer, to the supplement (Extended Data Figure 7) to provide a clearer illustration of these conclusions. This alignment includes all receptors used in our expanded selectivity analysis (which now includes GPR62 and GPR101, as noted elsewhere), as well as a subset of additional biogenic amine receptors.

- On page 13; line 292; *in silico* should written in italic.

- There seems to be an erroneous full stop on page 13; line 294.

- In some cases IC₅₀ is written as IC50, i.e. 50 should be in subscript.
- Page 27; line 501: A₂AR should be A₂AR (subscript)

We are grateful to the reviewer for pointing out each of these errors. The text has been amended in accordance with each of these points, as reproduced below:

Lines 314-316 (revised no markup version):

“To obtain structures of GPR61 in its inactive state, we utilized an efficient AlphaFold-driven *in silico* construct evaluation strategy to streamline the time-consuming process of experimental construct screening and optimization, providing significant savings in time and cost.”

Lines 548-549 (revised no markup version):

“The sequence inserted into GPR61 (comprising BRIL and A₂AR-derived linker sequences) is colored in cyan.”

- Page 40; line 728-732; some data seems to be missing here. Moreover the Y-axis of Figure 2d and 3e indicate that vehicle is set at 100% (while here it is state that vehicle is set at ZPE). The term ZPE and HPE are not used anywhere else in the manuscript.

The Y-axis of Figure 2d and 2e sets the basal cAMP activity of the construct at 100% activity, since the inhibition by compound 1 is normalized to the receptor's constitutive activity. In contrast, the meaning of zero percent effect (ZPE) and hundred percent effect (HPE) refers to the effect of the compound. ZPE is defined by the addition of vehicle in the absence of inhibitor, while HPE is defined by the addition of a high concentration of the non-specific inhibitor described above. Thus, the meaning of 100% cAMP activity for the receptor should be contrasted with the meaning of 100% effect (HPE) of the inverse agonist compounds. The existing text clarifies this, as follows:

Lines 801-803 (revised no markup version):

“For GPR61 WT and mutant profiling using transiently transfected cells, basal and inverse agonist effects were calculated as percent of the wild-type receptor signal measured in the presence of vehicle.”

Reviewer #3

This manuscript presents cryo-EM structures of both inactive and active states of GPR61, providing valuable molecular insights into the modulation of this receptor. The authors first determined the structure of GPR61 complexed with an engineered Gs heterotrimer, shedding light on the mechanism underlying the high constitutive activity of GPR61. Subsequently, they identified a novel inverse agonist, Compound 1, and elucidated the structures of inactive GPR61 alone and in complex with Compound 1. To achieve the cryo-EM structures of inactive GPR61, the authors utilized AlphaFold2 to guide their protein engineering strategy and designed a GPR61-BRIL fusion construct with continuous helicity at the junctions between TM5/6 and the BRIL helices. Notably, the structures of apo- and

Compound 1-bound GPR61 revealed a ligand-induced allosteric site for Compound 1, which has not been observed in other GPCRs. Based on the structures and mutagenesis studies, the authors further discussed the mechanism by which Compound 1 inhibits GPR61 signaling. Overall, this is an outstanding structural study on an orphan GPCR, with well-supported conclusions drawn from both structural and functional data. The cryo-EM structural models are well supported by their maps, especially for Compound 1. The figures presented are clear and informative, further enhancing the manuscript's impact.

We thank the reviewer for taking the time to examine our manuscript, as well as for returning comments on an accelerated timeline. We are also grateful for their kind words regarding the quality and significance of our findings. In the text below, we have attempted to address each of their concerns.

I only have some minor comments and issues:

1. The 'ionic lock' hypothesis was originally proposed based on structural studies of rhodopsin, which is required to remain completely inactive in the absence of light. The 'ionic lock' mechanism is thought to help maintain this inactive state by creating a stable interaction between the polar residues D3.50 and E6.30 in transmembrane helices 3 and 6, respectively. However, subsequent research has revealed that many Class A GPCRs do not possess this 'ionic lock' between D3.50 and E6.30 or other polar residues in TM6, indicating that it is not a universal mechanism for maintaining receptor inactivity in the absence of agonists. Therefore, lacking such 'ionic lock' may not be so special for GPR61 to explain its high basal activity.

We thank the reviewer for rightly pointing this out. We have amended the text to mention the fact that the ionic lock is not universally conserved:

Lines 113-119 (revised no markup version):

“The bias of GPR61 toward constitutive activation may not rely solely on its N-terminal peptide, however, as the following key sequence and structural features could be consistent with partial destabilization of the inactive state. For instance, in some (though not all) class A receptors, Arg^{3.50} of the conserved D/ERY motif participates in a salt bridge, called the “ionic lock”, with an acidic residue (Asp or Glu) in position 6.30, which creates an energetic barrier to the movement of TM6 to accommodate binding of the Gα C terminus during activation²⁵”

2. On the other hand, the distinctive disulfide bond formed between C6.47 and C7.45, located just above W6.48, is particularly intriguing. The microswitch residues W6.48 and F6.44 are typically involved in linking agonist binding to the outward displacement of TM6 in Class A GPCRs through conformational changes. It is possible that the C6.47 and C7.45 disulfide bond could somehow influence the conformation of W6.48, thereby facilitating the receptor's transition to the active state. Further mutagenesis studies targeting these two Cys residues or computational simulations could yield important insights into this potential mechanism.

We thoroughly agree with the reviewer's analysis. We have thus characterized a mutant version of the receptor (C299S) in which this disulfide is abolished to measure the effect of this mutation's basal cAMP activity. The results (reported in Extended Data Figure 1) indicate that disruption of this disulfide bond does not significantly impact the level of constitutive activity displayed by the receptor. As a result, we have modified the text to reflect these results, as follows:

Lines 131-134 (revised no markup version):

“To probe the function of this disulfide, we generated a GPR61 mutant (C299S) unable to form a disulfide bond, but found that it exhibits a similar level of basal activity to the wild-type receptor (Extended Data Figure 1), suggesting that it may not play a significant role in basal activation.”

3. Is it possible that the BRIL insertion somehow affects the conformation of TM6 causing the binding site of Compound 1 to be closed in the apo structure? Does the AlphaFold2 predicted structure have the allosteric site?

The AlphaFold 2 predictions both for the inactive wild-type version of GPR61 and for our BRIL construct lack the allosteric site, due to the inward-shifted position of TM6, as is commonly seen in other inactive-state receptors. While we cannot rule out the idea that the BRIL fusion has some influence on the positions of TM5 and TM6 in this construct, we clearly observe that the presence of Compound 1 induces the movement of the helices flanking the allosteric site, based on a comparison of the cryo-EM structures. It should be noted that the allosteric pocket induced by the compound also differs from the pocket occupied by the G protein in the complex, as detailed in Figure 4a.

Lines 336-340 (revised no markup version):

“While the artificial nature of the GPR61-BRIL fusion in GPR61_{IA} may give rise to concerns about whether the conformation of the apo protein is biologically relevant, the structural rearrangement induced by compound 1 suggests that the protein derived from this construct remains sufficiently flexible to accommodate binding of compound 1 in an induced pocket.”

4. Residues in the Disallowed Ramachandran Plot areas need to be adjusted for structures below 4Å resolution. Also, the clash score of the GPR61-G protein complex structure is too high.

We have further refined our models against the experimental maps to improve their statistics. Further refinement of the GPR61-G protein structure has significantly improved the overall statistics for the model, eliminated all Ramachandran outliers, and reduced the overall clashscore from 16.6 to 8.75. Similarly, further refinement of the GPR61-BRIL structure with compound 1 has been improved to eliminate any Ramachandran outliers. These improved statistics are detailed in the updated version of Extended Data Table 3 (now Extended Data Table 4). The updated coordinates for the active-state structure and the compound 1-bound structure of GPR61 have been deposited in the Protein Data Bank under codes 8TB0 and 8TB7.

5. The authors used CryoSparc to process their cryo-EM data. However, one may wonder if there was a specific reason for not employing heterogeneous refinement to identify more homogeneous particle subsets?

We agree with the reviewer's assessment that obtaining a homogeneous particle set requires a processing step, such as heterogeneous refinement, that allows sorting out of particle heterogeneity in 3D. In place of heterogeneous refinement, we have accomplished a similar effect by using "heterogeneous" *ab initio* modeling steps. In our hands, we found that *ab initio* modeling to generate multiple models using the parameters we described in our methods section (large subset sizes and a high resolution range) proved to be a more effective means of separating out the most homogeneous particle subset for the final model. This approach avoids any bias introduced by the choice of input models for heterogeneous refinement and ultimately gave us the best results.

REVIEWERS' COMMENTS

Reviewer #1 (Remarks to the Author):

The authors addressed my comments adequately and made the necessary revision on their manuscript. I strongly support the publication of the revised version in Nature Communications.

Reviewer #2 (Remarks to the Author):

This reviewer is satisfied with how feedback was incorporated. However, I feel strongly about the need for providing the chemical structure for the Pfizer compound, as this is integral to the conclusions of the paper.

Reviewer #3 (Remarks to the Author):

This is a very interesting paper from experts in the pharmaceutical industry. The authors have addressed all of my concerns.